# A novel method to derive the aerosol hygroscopicity parameter based only on measurements from a humidified nephelometer system

**Ye Kuang[1], ChunSheng Zhao[1], JiangChuan Tao[1], YuXuan Bian[2], Nan Ma[3], Gang Zhao[1]**

[1]{Department of Atmospheric and Oceanic Sciences, School of Physics, Peking University, Beijing, China}

[2]{State Key Laboratory of Severe Weather, Chinese Academy of Meteorological Sciences}

[3]{Leibniz Institute for Tropospheric research, Leipzig, Germany}

*Correspondence to: C. S. Zhao (zcs@pku.edu.cn)

**Abstract**

Aerosol hygroscopicity is crucial for understanding roles of aerosol particles in atmospheric chemistry and aerosol climate effects. Light scattering enhancement factor $f(\mathrm{RH}, \lambda)$ is one of the parameters describing aerosol hygroscopicity which is defined as $f(\mathrm{RH}, \lambda) = \sigma_{sp}(RH, \lambda)/\sigma_{sp}(dry, \lambda)$ where $\sigma_{sp}(RH, \lambda)$ or $\sigma_{sp}(dry, \lambda)$ represents $\sigma_{sp}$ at wavelength $\lambda$ under certain RH or dry conditions. Traditionally, an overall hygroscopicity parameter $\kappa$ can be retrieved from measured $f(\mathrm{RH}, \lambda)$, hereinafter referred to as $\kappa_{f(\mathrm{RH})}$, by combining concurrently measured particle number size distribution (PNSD) and mass concentration of black carbon. In this paper, a new method is proposed to directly derive $\kappa_{f(\mathrm{RH})}$ based only on measurements from a three-wavelength humidified nephelometer system. The advantage of this newly proposed approach is that $\kappa_{f(\mathrm{RH})}$ can be estimated without any additional information about PNSD and black carbon. This method is verified with measurements from two different field campaigns. Values of $\kappa_{f(\mathrm{RH})}$ estimated from this new method agree very well with those retrieved by using the traditional method, all points lie nearby 1:1 line, the square of correlation coefficient between them is 0.99. The verification results demonstrate that this newly proposed method of deriving $\kappa_{f(\mathrm{RH})}$ is applicable in different sites and seasons of the North China Plain and might be also applicable in other regions around the world.

## 1. Introduction

Atmospheric aerosol particles play vital roles in visibility, energy balance and the hydrological cycle of the Earth-atmosphere system and have attracted a lot of attention in recent decades. Aerosol particles suspended in the atmosphere directly influence radiative transfer of solar radiation and indirectly affect cloud properties, therefore, have large impacts on climate change. Especially, uncertainties in direct aerosol radiative forcing due to anthropogenic aerosols and in aerosol indirect forcing caused by aerosol interaction with clouds contribute most to the total uncertainty in climate forcing (Boucher et al., 2013). One of the most important factors affect these uncertainties is the interaction between aerosol particles and ambient atmospheric water vapour (Zhao et al., 2006;Kuang et al., 2016b). Under supersaturated conditions, aerosol particles serve as cloud condensation nuclei (CCN) and hence influence cloud properties. Under subsaturated conditions, with respect to typical aerosol compositions, water usually constitutes about half of the aerosol mass at a relative humidity (RH) of 80% with substantially higher water mass fractions existing at RH values above 90% for most ambient aerosol (Bian et al., 2014).The water content of aerosol and cloud droplets depend on both the ambient RH and hygroscopicity of the aerosol chemical constituents.

Traditionally, the Köhler theory (Petters and Kreidenweis, 2007) is widely used to describe the hygroscopic growth of aerosol particles and successfully used in laboratory studies for single component and some multicomponent particles. In order to account for the mixed organic and inorganic composition of ambient aerosol, Petters and Kreidenweis (2007) proposed a modified version of Köhler theory called κ-Köhler theory to describe a single aerosol hygroscopic growth parameter, κ. The κ-Köhler equation, expressed in terms of the diameter growth factor, g(RH), is given in equation (1) below:

$$\frac{RH}{100} = \frac{g^3-1}{g^3-(1-\kappa)} \cdot \exp\left(\frac{4\sigma_{s/a} \cdot M_{water}}{R \cdot T \cdot D_d \cdot g \cdot \rho_w}\right) \qquad (1)$$

where g corresponds to g(RH), $D_d$ is the dry diameter, $\sigma_{s/a}$ is the surface tension of solution/air interface, T is the temperature, $M_{water}$ is the molecular weight of water, R is the universal gas constant, $\rho_w$ is the density of water, and $\kappa$ is the hygroscopicity parameter. This theory is not only applicable to single-component aerosol particles, but also to multicomponent aerosol particles. With regard to a multicomponent aerosol particle, the Zdanovskii, Stokes, and Robinson assumption can be applied. The hygroscopicity parameter $\kappa$ of multicomponent aerosol particle can be derived by

using the following formula: $\kappa = \sum_i \varepsilon_i \cdot \kappa_i$, where $\kappa_i$ and $\varepsilon_i$ represent the hygroscopic parameter
and volume fraction of each component. This hygroscopicity parameter $\kappa$ has received much
attentions and turns out to be a very effective parameter to study aerosol hygroscopicity. This
hygroscopicity parameter $\kappa$ makes the comparison of the aerosol hygroscopicity at different sites
around the world and different time periods more convenient. In addition, hygroscopicity parameter
$\kappa$ also facilitates the intercomparison of aerosol hygroscopicity derived from different techniques and
measurements made at different RHs. This hygroscopicity parameter $\kappa$ is  widely used to account
the influence of aerosol hygroscopic growth on aerosol optical properties as well as aerosol liquid
water contents (Tao et al., 2014;Kuang et al., 2015;Brock et al., 2016;Bian et al., 2014;Zieger et al.,
2013) and to examine the role of aerosol hygroscopicty in CCN (Chen et al., 2014;Gunthe et al.,
2009;Ervens et al., 2010). The derived $\kappa$ values from field campaigns and laboratory studies will
further our understanding in aerosol hygroscopicity and help estimate the influences of aerosol
hygroscopic growth on different aspects of atmospheric processes.
The Humidity Tandem Differential mobility Analyzer (HTDMA) measures the aerosol diameter
hygroscopic growth as a function of RH. The aerosol hygroscopicity parameter $\kappa$ can be directly
derived from measurements of HTDMA by applying equation (1) (Liu et al., 2011;Wu et al., 2016).
HTDMA systems can    provide insights into the aerosol hygroscopicity at different aerosol diameters,
however, they can only be used to derive aerosol hygroscopicity parameter $\kappa$ within certain size range
(usually less than 300 nm). HTDMA systems are not capable of providing more details about aerosol
hygroscopicity of aerosol particles which contribute most to aerosol optical properties and aerosol
liquid water contents (their diameters usually ranging from 200 nm to 1μm) (Ma et al., 2012;Bian et
al., 2014). The effect of aerosol water uptake on the aerosol particle light scattering ($\sigma_{sp}$)   is usually
measured with a humidified nephelometer system. Measurements from a humidified nephelometer
system can also be used to calculate the aerosol hygroscopicty parameter $\kappa$ if the dry aerosol particle
number size distribution (PNSD) is measured simultaneously (Chen et al., 2014). The scattering
enhancement factor    $f(\mathrm{RH}, \lambda)$, defined as  $f(\mathrm{RH}, \lambda) = \sigma_{sp}(RH, \lambda)/\sigma_{sp}(dry, \lambda)$, characterizes
changes in the aerosol scattering coefficient with RH, $\sigma_{sp}(RH, \lambda)$ or $\sigma_{sp}(dry, \lambda)$ represents $\sigma_{sp}$ at
wavelength $\lambda$ at a certain RH or under dry conditions. In this research, $f(\mathrm{RH})$ is referred to as
$f(\mathrm{RH}, 550\ \mathrm{nm})$. The nephelometer measures aerosol optical properties of the entire aerosol size
distribution.    Thus, $\kappa$ calculated from $f(\mathrm{RH})$ measurements can be understood as an optically
weighted $\kappa$ and represents the overall hygroscopicty of ambient aerosol particles. This $\kappa$ is more
suitable for being used to account the influences of aerosol hygroscopic growth on aerosol optical
properties compared to aerosol hygroscopicity derived from HTDMA and CCN measurements.
Traditionally, derivation of κ from f(RH) measurements requires aerosol PNSD as well as black carbon
(BC) measurements to determine the imaginary part of the refractive index. As PNSD and BC
measurements are expensive, their availability in field campaigns are limited.

In this paper we use measurements from a field campaign on the North China Plain (NCP) to

derive κ values using 3 methods. The first 2 methods derive κ from aerosol diameter hygroscopic
growth and the third method derives an aerosol optical parameterization of κ. Method 1, labeled as
$\kappa_{f(RH)}$, derives κ from aerosol PNSD, BC and nephelometer f(RH) measurements. Method 2, defined
as $\kappa_{250}$, derives κ from g(RH) measurements of aerosol particles with diameter of 250 nm, using a
High-Humidity Differential Mobility Analyzer (HH-TDMA). HH-TDMA is a system very similar to
HTDMA but is capable of operating at higher RH points (Liu et al., 2011). Method 3, defined as $\kappa_{sca}$,
is an empirical determination of κ using only nepehelometer measurements of the aerosol scattering
coefficient as a function of RH.

Based on detailed analysis about the relationship between $\kappa_{f(RH)}$ and $\kappa_{sca}$, a novel method to

directly derive $\kappa_{f(RH)}$ based only on measurements from a humidified nephelometer system is
proposed. This newly proposed approach makes it more convenient and cheaper for researchers to
conduct aerosol hygroscopicity research with $f(RH)$ measurements.
**2.  Site description and instruments**

Datasets from five field campaigns are used in this paper. The five campaigns are conducted at

four different measurement sites of the North China Plain (NCP) (Wangdu, Xianghe and Gucheng in
Hebei province and Wuqing in Tianjin, and site locations are shown in Fig.S1). Time periods and used
datasets from these filed campaigns are listed in Table 1.

During these field campaigns, sampled aerosol particles have aerodynamic diameters less than 10

μm (selected by passing through an impactor). Aerosol PNSDs with particle diameter ranging from
3nm to 10μm were jointly measured by a Twin Differential Mobility Particle Sizer (TDMPS, Leibniz-
Institute for Tropospheric Research (IfT), Germany; Birmili et al. (1999)) or a scanning mobility
particle size spectrometer (SMPS) and an Aerodynamic Particle Sizer (APS, TSI Inc., Model 3321)

with a temporal resolution of 10 minutes. The mass concentrations of BC were measured using a Multi-angle Absorption Photometer (MAAP Model 5012, Thermo, Inc., Waltham, MA USA) or an aethalometer called AE33 (Drinovec et al., 2015). The aerosol light scattering coefficients ($\sigma_{sp}$) at three wavelengths were measured using a TSI 3563 nephelometer (Anderson and Ogren, 1998) or an Aurora 3000 nephelometer (Müller et al., 2011).

A humidified nephelometer system consists of two nephelometers and a humidifier was used in Wangdu and Gucheng campaigns. For the humidified nephelometer systems that we have designed, they only scan the hydration branch of the aerosol hygroscopic growth. The humidifier humidified the sample air through a Gore-Tex tube. The water vapor penetrates through the Gore-Tex tube, which is surrounded by a circulating water layer in a stainless steel tube. The temperature cycle of the circulating water layer was specified and controlled by a water bath. During Wangdu campaign, only one water bath was used, for each RH scanning cycle, the temperature cycle was fixed. Thus, the RH range of each cycle will change. Since the room temperature of the container was relatively stable during Wangdu campaign, the RH points of $f(\text{RH})$ cycles range from about 50% to about 90%, and each cycle lasted about 45 minutes. However, one cycle cost about 90 minutes because after each cycle was finished, the water bath needed about another 45 minutes to cool. During Gucheng campaign, this problem is solved by using two water baths and they provided circulating water alternatively for the humidifier. The corresponding temporal resolution of $f(\text{RH})$ cycles was about 45 minutes. In addition, a control software system was developed and can make sure the RH scans within certain RH range. During Gucheng campaign, the RH points of each $f(\text{RH})$ cycle range from 45% to 90%. During Wangdu campaign, the two nephelometers operated in series, used nephelometer was TSI 3563. During Gucheng campaign, the two nephelometers operated in parallel, used nephelometer was Aurora 3000. In the following, we refer the nephelometer which measures $\sigma_{sp}$ in dry state and the nephelometer which measures $\sigma_{sp}$ at different RH points as dry Neph and wet Neph, respectively. Two combined RH and temperature sensors (Vaisala HMP110; accuracy of $\pm0.2$ ℃ and $\pm1.7$ % for RH ranges from 0 to 90 %, respectively, and accuracy of $\pm2.5$ % for RH ranges from 90 % to 100 % according to the manufacturer) are placed at the inlet and outlet of the wet Neph, and the measured RHs and temperatures are defined as $RH_1/T_1$ and $RH_2/T_2$, respectively. The dew points at the inlet and outlet of wet Neph were calculated using the measured $RH_1/T_1$ and $RH_2/T_2$, and the average value was considered as the dew point of the sample air. The sample RH can be calculated

through the derived dew point and the sample temperature which is measured by the sensor inside the
sample cavity of the nephelometer. During Wangdu campaign, measurements from the humidified
nephelometer system were only available from 21 June, 2014, to 1 July , 2014. During the two
campaigns, the two nephelometers were calibrated every two weeks. The manufacturer of HMP110
suggests that the sensors should be calibrated yearly. We didn't calibrate the used HMP110 sensors
during the two campaigns because they only have been used less than three months, and results of
cross checks showed that they agree well with each other. The sample RHs in the dry Neph were about
20% and about 8% during Wangdu and Gucheng campaigns, respectively.

Dataset includes aerosol PNSDs at dry state, mass concentrations of BC and $\sigma_{sp}$ values of

different wavelengths from the following four campaigns which are listed in Table 1 are referred to as
dataset D1: two campaigns conducted in Wuqing, Xianghe campaign, Wangdu campaign before 21
June, 2014. Note that measured $\sigma_{sp}$ values of dataset D1 are not corrected for angular truncation
errors. This is because that dataset D1 is used for producing the look up table of the newly proposed
method, and it is expected that the Ångström exponent calculated from measured $\sigma_{sp}$ values can be
directly used as input for the newly proposed method. However, for $\sigma_{sp}$ values shown in Fig.1, the
angular truncation errors are corrected using Mie theory with measured PNSD and mass concentrations
of BC.

During Wangdu campaign, the growth factors of aerosol particles at six selected particle diameters

(30 nm, 50 nm, 100 nm, 150nm, 200 nm and 250 nm) at 98% RH condition were obtained from the
measurements of the HH-TDMA (Leibniz-Institute for Tropospheric Research (IfT), Germany; Hennig
et al. (2005)). Detailed information about HH-TDMA measurements please refer to Liu et al. (2011)
**3. Methodology**
**3.1 Calculations of hygroscopicity parameter $\kappa$ from $f(\mathrm{RH})$ measurements**

Research of Chen et al. (2014) demonstrated that if the PNSD at dry state is measured, then

measurements of $f(\mathrm{RH})$ can be used to derive the aerosol hygroscopicity parameter $\kappa$ by
conducting an iterative calculation with the Mie theory and the $\kappa$-Köhler theory. To reduce the
influence of random errors of observed $f(\mathrm{RH})$ at a certain RH, all valid $f(\mathrm{RH})$ measurements in a
complete humidifying cycle is used in the derivation algorithm. The retrieved $\kappa$ is the $\kappa$ value which
can be used to best fit the observed $f(\mathrm{RH})$ curve, labelled as $\kappa_{f(RH)}$, and this method of deriving $\kappa$

is Method 1. Details about this retrieval algorithm is described in Chen et al. (2014). Of particular note is that in this research the mass concentration of BC is also considered in the retrieval algorithm to account for the influence of BC on refractive indices of aerosol particles at different sizes. During the simulating process, aerosol components are divided into two classes in terms of their optical properties: the light absorbing component (i.e. BC) and less absorbing components (comprising inorganic salts and acids such as sulfates, nitrates, ammoniums, as well as most of the organic compounds). The BC is considered to be homogeneously mixed with other aerosol components, and the mass size distribution of BC used in Ma et al. (2012) which is observed on the NCP is used in this research to account the mass distributions of BC at different particle sizes. The used refractive index and density of BC are $1.80 - 0.54i$ and $1.5\mathrm{g}\ cm^{-3}$ (Kuang et al., 2015). Used refractive indices of non light-absorbing aerosol components (other than BC) and liquid water are $1.53 - 10^{-7}i$ (Wex et al., 2002) and $1.33 - 10^{-7}i$ (Seinfeld and Pandis, 2006), respectively. The flow chart about this retrieval algorithm is also introduced in the supporting information, please refer to Fig.S2 for more details.

**3.2 Calculations of hygroscopicity parameter $\kappa$ from HH-TDMA measurements**

The HH-TDMA measures hygroscopic growth factors of particles at different sizes at 98% RH condition. The measured hygroscopic factors can be directly related to $\kappa$ with equation (1). For a specified size of selected aerosol particles, a distribution of growth factors can be measured, and thus can be used to derive a probability distribution of $\kappa$ and finally come to the calculation of average $\kappa$ value corresponding to this size of aerosol particles. The method on how to derive average $\kappa$ value of certain size of aerosol particles from HH-TDMA measurements is elaborately described in Liu et al. (2011). In this research $\kappa$ values derived from g(RH) measurements of aerosol particles with diameter of 250 nm are used, defined as $\kappa_{250}$. This method of deriving $\kappa$ is Method 2.

**3.3 Parameterization schemes for $f(\mathrm{RH})$**

The most frequently used $f(\mathrm{RH})$ parameterization scheme is a power-law function which is known as "gamma" parameterization (Hänel, 1981) and the formula of this single-parameter representation is written as the following:

$$f(\mathrm{RH}) = \left[\frac{100-RH_0}{100-RH}\right]^{\gamma} \quad (2)$$

where $RH_0$ is the RH of dry condition, and $\gamma$ is a parameter fitted to the observed $f(\mathrm{RH})$. In this study, we estimated $\gamma$ values with observed $f(\mathrm{RH})$ curves and for the first time to our knowledge,

we further examined the relationship between $\gamma$ and $\kappa_{f(RH)}$.
Recently, a new physically based single-parameter representation was proposed by Brock et al.
(2016) to describe $f(RH)$. Their results demonstrated that this proposed parameterization scheme can
better describe $f(RH)$ than the widely used gamma power-law approximation (Brock et al., 2016).
The formula of this new scheme is written as:
$$f(RH) = 1 + \kappa_{sca}\frac{RH}{100-RH} \qquad (3)$$
where $\kappa_{sca}$ is a parameter fits $f(RH)$ best. Regardless of the curvature effects for particle diameters
larger than 100 nm, the hygroscopic growth factor for aerosol particles can be approximately expressed
as the following (Brock et al., 2016): $gf_{diam} \cong (1 + \kappa\frac{RH}{100-RH})^{1/3}$. Moreover, $\sigma_{sp}$ is usually
approximately proportional to total aerosol volume (Pinnick et al., 1980) which means that the relative
change in $\sigma_{sp}$ due to aerosol water uptake is roughly proportional to relative change in aerosol volume.
The enhancement factor in volume can be expressed as the cube of $gf_{diam}$, thus lead to the formula
form of $f(RH)$ expressed in equation (3).
During processes of measuring $f(RH)$, the sample RH in the dry Neph ($RH_0$) is not zero.
According to equation (3), the measured $f(RH)_{measure} = \frac{f(RH)}{f(RH_0)}$ should be fitted using the following
formula:
$$f(RH)_{measure} = (1 + \kappa_{sca}\frac{RH}{100-RH})\big/(1 + \kappa_{sca}\frac{RH_0}{100-RH_0}) \qquad (4)$$
The method of calculating $\kappa_{sca}$ by curve fitting using equation (4) is called Method 3. The
"gamma" parameterization scheme is referred to as $\gamma$-Method in the following paragraphs.
**4.  Results and discussions**
**4.1 Derived $\kappa$ values from $f(RH)$ and HH-TDMA measurements**
During this field campaign, the aerosol physical, chemical and optical properties are
synergistically observed with different types of instruments. They provide valuable datasets to perform
an insightful analysis about aerosol hygroscopicity and its relationship with other aerosol properties.
The time series of $\sigma_{sp}$ at 550 nm at dry state are shown in Fig.1a.. The results show that this
observation period has experienced varying degrees of pollution levels, with $\sigma_{sp}$ at 550 nm ranging
from 15 to 1150 $Mm^{-1}$.Values of $\kappa_{f(RH)}$ derived from Method 1 are shown in Fig.1b. During
deliquescence f(RH) exhibits an abrupt increase between RH values of 60-65%. As such, only f(RH)

data points with RH >70% were used in determination of $\kappa_{f(RH)}$ when deliquescence was apparent. For f(RH) cycles without deliquescence, all f(RH) points are used in the retrieval algorithm with RH ranges of about 50% to 90%. The results demonstrate that $\kappa_{f(RH)}$ lies between 0.06 and 0.51, with an average of 0.32. The lowest $\kappa_{f(RH)}$ values are found when the air quality is relatively clean ($\sigma_{sp}$ at 550 nm is lower than 100 $Mm^{-1}$) on 27 and 28 June. During these two days, organic matter dominates the mass concentration of PM2.5 which results in the low hygroscopicity of aerosol particles (Kuang et al., 2016a). On the contrary, the largest $\kappa_{f(RH)}$ values are found during periods when deliquescent phenomena occur and inorganic chemical compositions dominate the mass concentrations of PM2.5, especially, sulfate is highly abundant during these periods. Of particular note is that during relatively polluted periods ($\sigma_{sp}$ at 550 nm larger than 100 $Mm^{-1}$) aerosol particles are generally very hygroscopic which imply that aerosol water uptake can exert significant impacts on regional direct aerosol radiative effect and ambient visibility during this observation period.

On the basis of the average size-resolved $\kappa$ distribution from Haze in China (HaChi) campaign (Liu et al., 2014), $\kappa$ values change a lot for aerosol particles whose diameters are less than 250 nm, however, $\kappa$ values vary relatively smaller for aerosol particles whose diameter range from 250 nm to 1 μm. In addition, the results from HaChi campaign also demonstrate that aerosol particles whose diameter range from 200 nm to 1 μm usually contribute more than 80% to $\sigma_{sp}$ at 550 nm during summer on the NCP (Ma et al., 2012). That is, $\kappa_{f(RH)}$ may share similar magnitude with $\kappa_{250}$. To compare $\kappa$ values derived from Method 1 and Method 2values of $\kappa_{250}$ are also shown in Fig.1b. During this observation period, values of $\kappa_{250}$ range from 0.11 to 0.56, with an average of 0.34 which is very close to average $\kappa_{250}$ observed during HaChi campaign (Liu et al., 2011). The results shown in Fig.1b suggest that, in general, $\kappa_{f(RH)}$ values agree well with $\kappa_{250}$ values, however, are usually lower than $\kappa_{250}$ values. To quantitatively compare these two types of $\kappa$ values, they are plotted against each other and shown in Fig.2. It can be seen that they are highly correlated but overall, the $\kappa_{250}$ values are higher than $\kappa_{f(RH)}$ values, and the average difference between $\kappa_{250}$ and $\kappa_{f(RH)}$ is 0.02. The statistical relationship between $\kappa_{250}$ and $\kappa_{f(RH)}$ is also shown in Fig.2.This relationship may be useful for researchers if they want to estimate the influences of aerosol water uptake on aerosol optical properties and aerosol liquid water contents when only HH-TDMA or HTDMA measurements are available.

A model experiment is conducted to better understand the relationship between $\kappa_{250}$ and $\kappa_{f(\mathrm{RH})}$.
During HaChi campaign, size-resolved $\kappa$ distributions are derived from measured size-segregated
chemical compositions (Liu et al., 2014) and their average is used in this experiment to account the
size dependence of aerosol hygroscopicity which is shown in Fig. 3a. With this fixed average size-
resolved $\kappa$ distribution, all observed PNSDs at dry state along with mass concentrations of BC from
dataset D1 are used to simulate the retrieval of $\kappa_{f(\mathrm{RH})}$ under different PNSD and BC conditions. The
used PNSDs shown in Fig.3b indicate that large varying types of PNSDs are considered in the
simulative experiment. As to the simulating process, with given PNSD, mass concentration of BC and
size-resolved $\kappa$ distribution. The first step is simulating $f(\mathrm{RH})$ points using Mie theory and κ-
Köhler theory with RH range of 50% to 90% and the RH interval is 10%. The second step is retrieving
corresponding $\kappa_{f(\mathrm{RH})}$ using the procedure of Method 1. The $\kappa$ value at particle diameter of 250 nm
of the used size-resolved $\kappa$ distribution is the corresponding $\kappa_{250}$. The probability distribution of
simulated $\kappa_{f(\mathrm{RH})}$ is also shown in Fig.3a. The standard deviation of retrieved $\kappa_{f(\mathrm{RH})}$ is about 0.01
which suggests that if the size-resolved $\kappa$ distribution is fixed, then $\kappa_{f(\mathrm{RH})}$ varies little. Due to
$\kappa_{f(\mathrm{RH})}$ represents an overall, size-integrated $\kappa$, it is clearly shown in Fig.3a that in most cases $\kappa_{f(\mathrm{RH})}$
values are located between $\kappa$ values of aerosol particles ranging from 200 nm to 1μm. Moreover,
about 70% of simulated $\kappa_{f(\mathrm{RH})}$ values are less than $\kappa_{250}$ which to some extent explains the observed
difference between $\kappa_{250}$ and $\kappa_{f(\mathrm{RH})}$ mentioned before. However, the simulated average difference
between $\kappa_{250}$ and average $\kappa_{f(\mathrm{RH})}$ is about 0.01 which is less than the observed averaged difference
between $\kappa_{250}$ and $\kappa_{f(\mathrm{RH})}$ which is 0.02. Especially, when $\kappa_{f(\mathrm{RH})}$ values are relatively lower ($<$
0.25), the $\kappa_{250}$ is systematically higher than $\kappa_{f(\mathrm{RH})}$. Except that uncertainties from measurements of
instruments, for example, the uncertainty of RH in measurements of HH-TDMA and uncertainties of
measuring $f(\mathrm{RH})$ (details about the uncertainty sources of $f(\mathrm{RH})$ measurements can be found in
the paper published by Titos et al. (2016)), there are other two reasons may be associated with the
discrepancy between $\kappa_{250}$ and $\kappa_{f(\mathrm{RH})}$. The first one is that configurations of size-resolved $\kappa$
distributions and PNSDs during this field campaign are far different from the model experiment. The
second one is that in the real atmosphere, $\kappa$ values at different RH conditions may be different (You
et al., 2014) and most of $f(\mathrm{RH})$ measurements are conducted when RH is lower than 90%, however,
the measurements of HH-TDMA are conducted when RH is equal to 98% . Overall, the observed
general consistency between $\kappa$ values derived from measurements of $f(\mathrm{RH})$ and HH-TDMA
confirms the reliability of $\kappa$ values derived from $f(\text{RH})$ measurements.
**4.2 Relationships between $\kappa$ derived from $f(\text{RH})$ measurements and $f(\text{RH})$ fitting**
**parameters**
In the previous section, derived $\kappa_{f(\text{RH})}$ values are characterized and compared with $\kappa_{250}$ values.
These results demonstrated that derived $\kappa_{f(\text{RH})}$ values can commendably represent variations of
aerosol hygroscopicty of ambient aerosol populations. In this section, the relationship between derived
$\kappa_{f(\text{RH})}$ values and $f(\text{RH})$ fitting parameters are further examined to investigate their relationships.
Two parameterization schemes of $f(\text{RH})$ are discussed in this paper, including the $\gamma$-Method
and Method 3. Values of $\gamma$ and $\kappa_{sca}$ are fitted from observed $f(\text{RH})$ cycles. For cycles during
deliquescent periods, only $f(\text{RH})$ points with RH higher than 70% are used to perform fitting
processes. The relationship between $\kappa_{f(\text{RH})}$ and $\gamma$ is investigated and shown in Fig.4a. It is found
that an approximately linear relationship exists (square of correlation coefficient is 0.90) between
$\kappa_{f(\text{RH})}$ and $\gamma$, especially when $\kappa_{f(\text{RH})}$ is larger than 0.2. During this field campaign, fitted $\gamma$ ranges
from 0.13 to 0.56 with an average of 0.41.
During this field campaign, fitted $\kappa_{sca}$ ranges from 0.05 to 0.36 with an average of 0.22. The
relationship between $\kappa_{f(\text{RH})}$ and $\kappa_{sca}$ is also investigated and shown in Fig.4b. It is found that a
strong linear relationship also exists (square of correlation coefficient is 0.97) between $\kappa_{f(\text{RH})}$ and
$\kappa_{sca}$. The statistically fitted line almost passes though zero point which implies that a proportional
relationship may exist between $\kappa_{f(\text{RH})}$ and $\kappa_{sca}$. This strong correlation should be intrinsic due to
the idea of Method 3 is from the linkage between total aerosol volume and $\sigma_{sp}$ as introduced in
Sect.3.3 and the increase of total aerosol volume due to aerosol water uptake is directly linked to the
overall aerosol hygroscopicity parameter $\kappa$. It seems that this promising linear relationship can help
bridge the gap between $f(\text{RH})$ and $\kappa$. However, results from Brock et al. (2016) imply that the
relationship between $\kappa_{f(\text{RH})}$ and $\kappa_{sca}$ is much more sophisticated and it is affected by both aerosol
hygroscopicity and PNSD at dry state. In the paper published by Brock et al. (2016), $\kappa_{ext}$ (a parameter
determined from measurements of the aerosol extinction coefficient as a function of RH using formula
form of equation (2)) and $\kappa_{chem}$ (a constant $\kappa$ determined from chemical constituents of entire
aerosol population) are used and correspond to $\kappa_{sca}$ and $\kappa_{f(\text{RH})}$ in this research, the difference
between $\kappa_{ext}$ and $\kappa_{sca}$ is that $\kappa_{ext}$ is used to fit the light enhancement factor of aerosol extinction
coefficient, $\kappa_{chem}$ and $\kappa_{f(RH)}$ actually means the same because both them are overall and size
independent hygroscopicity parameters. Results from Brock et al. (2016) concluded that the ratio
$\kappa_{ext}/\kappa_{chem}$ generally lies between 0.6 to 1 which implies that the ratio $\kappa_{sca}/\kappa_{f(RH)}$ (in the following,
this ratio is referred to as $R_\kappa$) also should have large variations and may share a similar range of
variability. By revisiting the relationship between $\kappa_{f(RH)}$ and $\kappa_{sca}$ found in this research, it can be
found that $R_\kappa$ during this field campaign ranges from 0.58 to 0.77 with an average of 0.69. This result
suggests that if we directly establish a linkage between $\kappa_{f(RH)}$ and $\kappa_{sca}$ with an average $R_\kappa$ can
result in a non-negligible bias (relative difference can reach about 15%). Besides, this range of $R_\kappa$
only represents the relationship between $\kappa_{f(RH)}$ and $\kappa_{sca}$ during a short time period and at only one
site.

To better understand the relationship between $\kappa_{f(RH)}$ and $\kappa_{sca}$, all PNSDs at dry state (shown

in Fig.3a) along with mass concentrations of BC from dataset D1 are used to simulate the relationship
between $\kappa_{f(RH)}$ and $\kappa_{sca}$ with Mie and κ-Köhler theories. The aim of including PNSD and BC
information from different campaigns is to simulate variations of $R_k$ under different conditions.
During simulating processes, for each PNSD, we change $\kappa_{f(RH)}$ from 0.01 to 0.7 with an interval of
0.01 to examine the influence of aerosol hygroscopicity on $R_\kappa$. For each PNSD and $\kappa_{f(RH)}$, the
simulating processes include two steps. The first step is simulating $f(RH)$ points using Mie theory
and κ-Köhler theory with RH range of 50% to 90% and the RH interval is 10%. The way of treating
BC is same with the retrieval procedure of $\kappa_{f(RH)}$ introduced in Sect.3.1. The second step is retrieving
the corresponding $\kappa_{f(RH)}$ using the procedure of Method 1 and calculating $\kappa_{sca}$ with Method 3.
Simulated results of $\kappa_{f(RH)}$ and $\kappa_{sca}$ are shown in Fig.5a and the probability distribution of
corresponding $R_\kappa$ values is shown in Fig.5b. The results show that $R_\kappa$ primarily ranges from 0.5 to
0.84 with an average of 0.69 which is same with the average $R_\kappa$ measured during Wangdu campaign.
These results also indicate that the relationship between $\kappa_{f(RH)}$ and $\kappa_{sca}$ is much more complex than
a simple linear relationship and more information about aerosol properties are necessary to gain
insights into the variation of $R_\kappa$.
**4.3 A novel method to directly derive $\kappa$ from measurements of a humidified nephelometer**
**system**

A robust linear relationship is   found between $\kappa_{f(RH)}$ and $\kappa_{sca}$ in Sect.4.2 , however, results

of further analysis suggest that $R_\kappa$ varies a lot. The complexity comes from that both PNSD at dry

state and aerosol hygroscopicity have impacts on $R_\kappa$. Generally, used nephelometer of a humidified

nephelometer system have three wavelengths (Titos et al., 2016) and the spectral dependence of $\sigma_{sp}$

is usually described by the following Ångström formula: $\sigma_{sp}(\lambda) = \beta\lambda^{-\alpha_{sp}}$, where $\beta$ is the particle

number concentration dependent coefficient, $\lambda$ is the wavelength of light and $\alpha_{sp}$ represents the

Ångström exponent of $\sigma_{sp}$. Ångström exponent can be directly inferred from the measurements

of $\sigma_{sp}$ at different wavelengths. Of particular note is that Ångström exponent not only can be used

to account the spectral course of $\sigma_{sp}$, it also reveals information about PNSD. In general, larger value

of Ångström exponent corresponds to smaller aerosol particles. That is, Ångström exponent can

be a proxy of PNSD at dry state and be used in the processes of estimating the impacts of PNSD on

$R_\kappa$. On the other hand, with regard to aerosol hygroscopicity, although $R_\kappa$ varies within certain range,

value of $\kappa_{sca}$ can still be used to represent the overall hygroscopicity of aerosol particles.

Simulated $R_\kappa$ values introduced in the last paragraph of Sect.4.2 are spread into a two

dimensional gridded plot, they are simulated from all observed PNSDs and mass concentrations of BC

of dataset D1. The first dimension is Ångström exponent with an interval of 0.02 and the second

dimension is $\kappa_{sca}$ with an interval of 0.01, average $R_\kappa$ value within each grid is represented by color

and shown in Fig.6a. Values of Ångström exponent corresponding different PNSDs are calculated

from concurrently measured $\sigma_{sp}$ values at 450 nm and 550 nm from TSI 3563 nephelometer. Based

on results shown on Fig.6a, the different impacts of aerosol hygroscopicity and dry scattering

Ångström exponent on $R_\kappa$ can be distinguished to some extent. The results demonstrate that PNSD

at dry state play a more important role in the variations of $R_\kappa$ than overall aerosol hygroscopicty..

Overall, larger value of Ångström exponent corresponds to higher $R_\kappa$. Aerosol hygroscopicty

exhibits different influences on $R_\kappa$ when Ångström exponent values are different. On average,

higher $\kappa_{sca}$ corresponds to lower $R_\kappa$ if Ångström exponent is smaller than about 0.8 and higher

$\kappa_{sca}$ corresponds to higher $R_\kappa$ if Ångström exponent is larger than about 1.6.The percentile value

of standard deviation of $R_\kappa$ values within each grid of Fig.6a divided by their average is shown in

Fig.6b. The Ångström exponent only represents an overall size property of aerosol particles, the

same Ångström exponent corresponds to different aerosol PNSDs. Within each grid of Fig.6a, the

same $\kappa_{sca}$ corresponds to different combinations of aerosol PNSD and $\kappa_{f(\mathrm{RH})}$, and $R_\kappa$ values also

change. Note that the size-dependent chemical composition also exerts influence on $R_k$. However, if

PNSD is fixed, each size-resolved $\kappa$ distribution corresponds to a certain $\kappa_{f(RH)}$, and $\kappa_{f(RH)}$ varies
with certain range no matter how size-resolved $\kappa$ distribution changes. Therefore, influences of size-
dependent chemical compositions on $R_k$ are already included in the simulated results of producing
the look up table by varying the $\kappa_{f(RH)}$ from 0 to 0.7 for a fixed aerosol PNSD.
As shown in Fig.6b, in most cases, these percentile values are less than 6% (about 80%) which
demonstrates that $R_\kappa$ varies little within each grid shown in Fig.6a. This implies that results of Fig.6a
can be used as a look up table to estimate $R_\kappa$. As what's introduced before, currently used
nephelometer of a humidified nephelometer system usually have three wavelengths (Titos et al., 2016),
thus can provide information about Ångström exponent, and $\kappa_{sca}$ can be directly fitted from
observed $f(RH)$ curve. Even only one $f(RH)$ point is measured, $\kappa_{sca}$ can still be calculated from
equation (4). Using results shown in Fig.6a as a look up table, $R_\kappa$ values can be directly predicted
from measurements of a humidified nephelometer system. With this method, $R_\kappa$ values during this
Wangdu field campaign are predicted (values of Ångström exponent are calculated from measured
$\sigma_{sp}$ values at 450 nm and 550 nm under dry conditions) and compared with measured $R_\kappa$ values, the
results are shown in Fig.7a. The Ångström exponent during this field campaign ranges from 0.63 to
1.96 with an average of 1.4. It can be seen from Fig.7a that majority of points lie nearby 1:1 line and
82% points have relative differences less than 6% which is consistent with results shown in Fig.6b.
This result is quite promising and can be further used to derive $\kappa_{f(RH)}$ values by combining fitted
$\kappa_{sca}$ and predicted $R_\kappa$. This method of deriving $\kappa_{f(RH)}$ is called Method 4 and include two steps.
The first step is calculating Ångström exponent based on measured $\sigma_{sp}$ values at 450 nm and 550
nm by the dry nephelometer and calculating $\kappa_{sca}$ based on measured $f(RH)$ curve by the humidified
nephelometer system. The second step is predicting $R_\kappa$ using the look up table shown in Fig.6a, and
then calculate $\kappa_{f(RH)}$ based on predicted $R_\kappa$ and fitted $\kappa_{sca}$. The results of predicted $\kappa_{f(RH)}$ values
are shown in Fig.7b and a robust correlation between $\kappa_{f(RH)}$ values predicted from Method 4 and
$\kappa_{f(RH)}$ values derived from Method 1 is achieved (the square of correlation coefficient between them
is 0.99). All points shown in Fig.7b lie nearby 1:1 line, average difference between $\kappa_{f(RH)}$ derived
from Method 4 and Method 1 is -0.009.
Datasets from Gucheng campaign are further used to verify Method 4. In this campaign, Aurora
3000 nephelometer is used for the humidified nephelometer system, it has three wavelengths: 450 nm,

525 nm and 635 nm. Values of $f(\mathrm{RH}, \lambda)$ points correspond to wavelength of 525 nm are used to derive $\kappa_{f(\mathrm{RH})}$ using Method 1 and Method 4, used RH range is 45% to 90%. The look up table shown in Fig.6a is simulated corresponding to scattering wavelength of 550 nm, and is not suitable for being used in Method 4 if the nephelometer is Aurora 3000. A new look up table is simulated corresponding to scattering wavelength of 525 nm, used datasets of PNSD and BC are same with those for producing the look up table shown in Fig.6a. During Gucheng campaign, the variations of $\kappa_{f(\mathrm{RH})}$ and corresponding $R_k$ with $\sigma_{sp}$ at 525 nm are shown in Fig.8a. Values of $\kappa_{f(\mathrm{RH})}$ range from 0.01 to 0.27, with an average of 0.14. During this campaign, $\kappa_{f(\mathrm{RH})}$ is relatively lower when $\sigma_{sp}$ is high. Values of $R_k$ range from 0.60 to 0.84, with an average of 0.7. Results of the comparison between $\kappa_{f(\mathrm{RH})}$ derived from Method 1 and Method 4 are shown in Fig.8b. The results demonstrate that good consistency is achieved between $\kappa_{f(\mathrm{RH})}$ derived from Method 1 and Method 4, the square of correlation coefficient between them is 0.99.

The verification results of Method 4 using measurements from Wangdu and Gucheng campaigns demonstrate that a quite good estimation of $\kappa_{f(\mathrm{RH})}$ can be achieved by using only measurements from a humidified nephelometer system. The processes of simulating the look up table are independent of the size-resolved $\kappa$ distribution, and used PNSDs are from four different field campaigns which were conducted in different sites and seasons of the NCP. The verification datasets from two different field campaigns are totally independent of the look up table and also from different sites and seasons of the NCP. These results demonstrate that the newly proposed method is applicable in different sites and seasons of the NCP. The results shown in Fig.6b demonstrate that if Ångström exponent and $\kappa_{sca}$ are fixed, then $R_k$ varies little. The maximum $\kappa_{sca}$ of the look up table is 0.4, if $R_k$ is 0.8 (close to the simulated highest $R_k$ shown in Fig.5b), the corresponding $f(\mathrm{RH} = 80\%)$ is 2.6. According to the review of Titos et al. (2016), most of $f(\mathrm{RH} = 80\%)$ values for continental aerosols are lower than 2.6. The Ångström exponent range of the look up table is 0.4 to 2.0. Which demonstrate that the look up table shown in Fig.6a already covers large variation ranges of Ångström exponent and $\kappa_{sca}$ and can be used under different conditions. Thus, the newly proposed method of deriving $\kappa_{f(\mathrm{RH})}$ might be also applicable in other regions around the world. However, it should be pointed out that the look up table shown in Fig.6a is from simulations of measured continental aerosols without influences of desert dust, and it might not be suitable for being used to estimate $\kappa_{f(\mathrm{RH})}$ when sea salt or dust particles prevail. In summary, this approach allows researchers to directly derive aerosol

hygroscopicity from measurements of $f(\text{RH})$ without any additional information about PNSD and
BC which is quite convenient for researchers to conduct aerosol hygroscopicity researches with
measurements from a humidified nephelometer system.

**5.  Conclusions**

In this paper, values of aerosol hygroscopicity parameter $\kappa$ during Wangdu campaign are first
derived from measurements of $f(\text{RH})$ by combining measurements of PNSD at dry state and BC.
The results show that during this field campaign, aerosol hygroscopicity varies a lot, and $\kappa_{f(\text{RH})}$
ranges from 0.06 to 0.51 with an average of 0.34. Retrieved $\kappa_{f(\text{RH})}$ values are further compared with
$\kappa_{250}$ which is derived from measurements of HH-TDMA and good consistency is achieved.
Relationships between $\kappa_{f(\text{RH})}$ and $f(\text{RH})$ fitting parameters $\gamma$ and $\kappa_{sca}$ are further
investigated in Sect.4.2 which is for the first time to our knowledge. Good linear relationship is found
between $\kappa_{f(\text{RH})}$ and $\kappa_{sca}$ during Wangdu campaign. Results of detailed analysis about the
relationship between $\kappa_{f(\text{RH})}$ and $\kappa_{sca}$ demonstrate that the relationship between $\kappa_{f(\text{RH})}$ and $\kappa_{sca}$
is complicated, and the ratio $\kappa_{sca}/\kappa_{f(\text{RH})}$ $(R_\kappa)$ varies a lot (0.5 to 0.84, with an average of 0.69).
In Sect.4.3, a look up table based on Ångström exponent and $\kappa_{sca}$ is developed to estimate $R_\kappa$.
With this look up table, $R_\kappa$ as well as $\kappa_{f(\text{RH})}$ can be directly estimated from measurements of a
humidified nephelometer system. This method is further verified with measurements from two
different campaigns. The verification results demonstrate that a quite good estimation of $\kappa_{f(\text{RH})}$ can
be achieved by using only measurements from a humidified nephelometer system, and this method is
applicable at different sites and in different seasons of the NCP and might be also applicable in other
regions around the world. This newly proposed novel approach allow researchers to estimate $\kappa_{f(\text{RH})}$
without any additional information about PNSD and BC. This new finding directly links $\kappa$ and
$f(\text{RH})$ and will make the humidified nephelometer system more convenient when it comes to aerosol
hygroscopicity research. Finally, findings in this research may facilitate the intercomparison of aerosol
hygroscopicity derived from different techniques, help for parameterizing $f(\text{RH})$ and predicting
CCN properties with optical measurements.


**Acknowledgments**

This work is supported by the National Natural Science Foundation of China (41590872, 41375134).
The data used are listed in the references and a repository at http://pan.baidu.com/s/1c2Nzc5a.

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

**Table 1**. Locations, time periods and used datasets of five field campaigns

| Location | Wuqing | Wuqing | Xianghe | Wangdu | Gucheng |
|---|---|---|---|---|---|
| Time period | 7 march to 4 April, 2009 | 12 July to 14 August, 2009 | 9 July to 8 August, 2013 | 4 June to 14 July, 2014 | 15 October to 25 November, 2016 |
| PNSD | TSMPS+APS | TSMPS+APS | TSMPS+APS | TSMPS+APS | SMPS+APS |
| BC | MAAP | MAAP | MAAP | MAAP | AE33 |
| $\sigma_{sp}$ | TSI 3563 | TSI 3563 | TSI 3563 | TSI 3563 | Aurora 3000 |
| $f(\mathrm{RH})$ | | | | Humidified nephelometer system | Humidified nephelometer system |
| g(RH) | | | | HH-TDMA | |


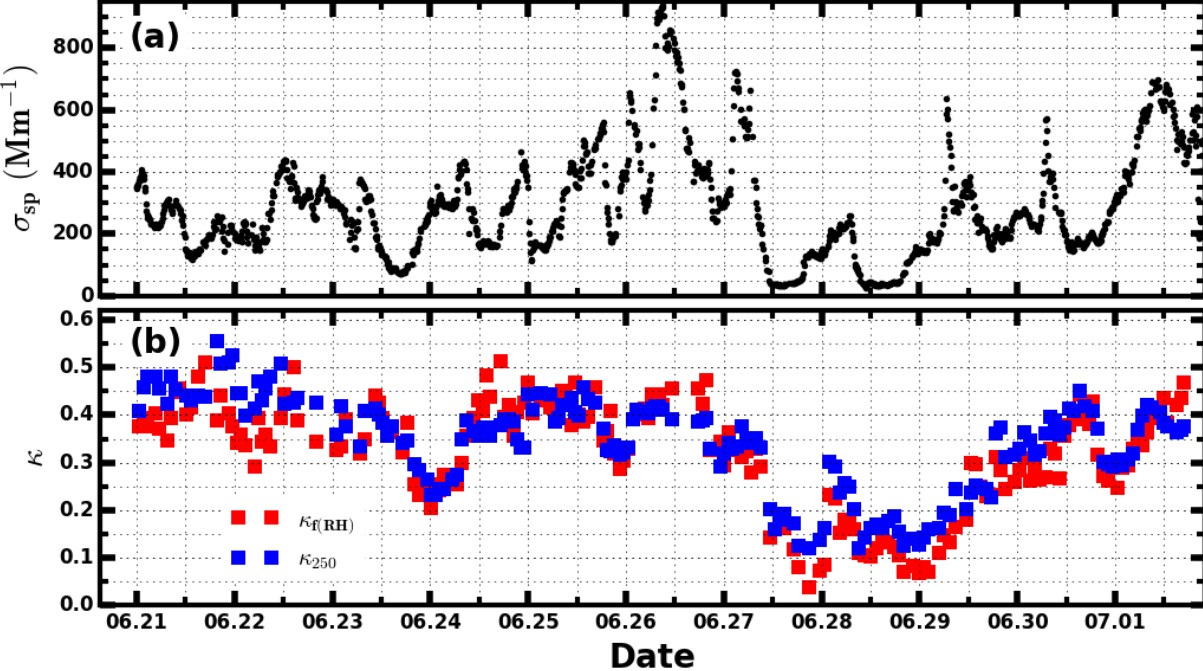


**Figure 1**. (a) The time series of $\sigma_{sp}$ at 550 nm; (b) The time series of $\kappa$ values derived from $f(\mathrm{RH})$ measurements ($\kappa_{f(\mathrm{RH})}$) by combining information of PNSD and BC, and time series of average $\kappa$ values of aerosol particles at 250 nm ($\kappa_{250}$) which is calculated from measurements of HH-TDMA.


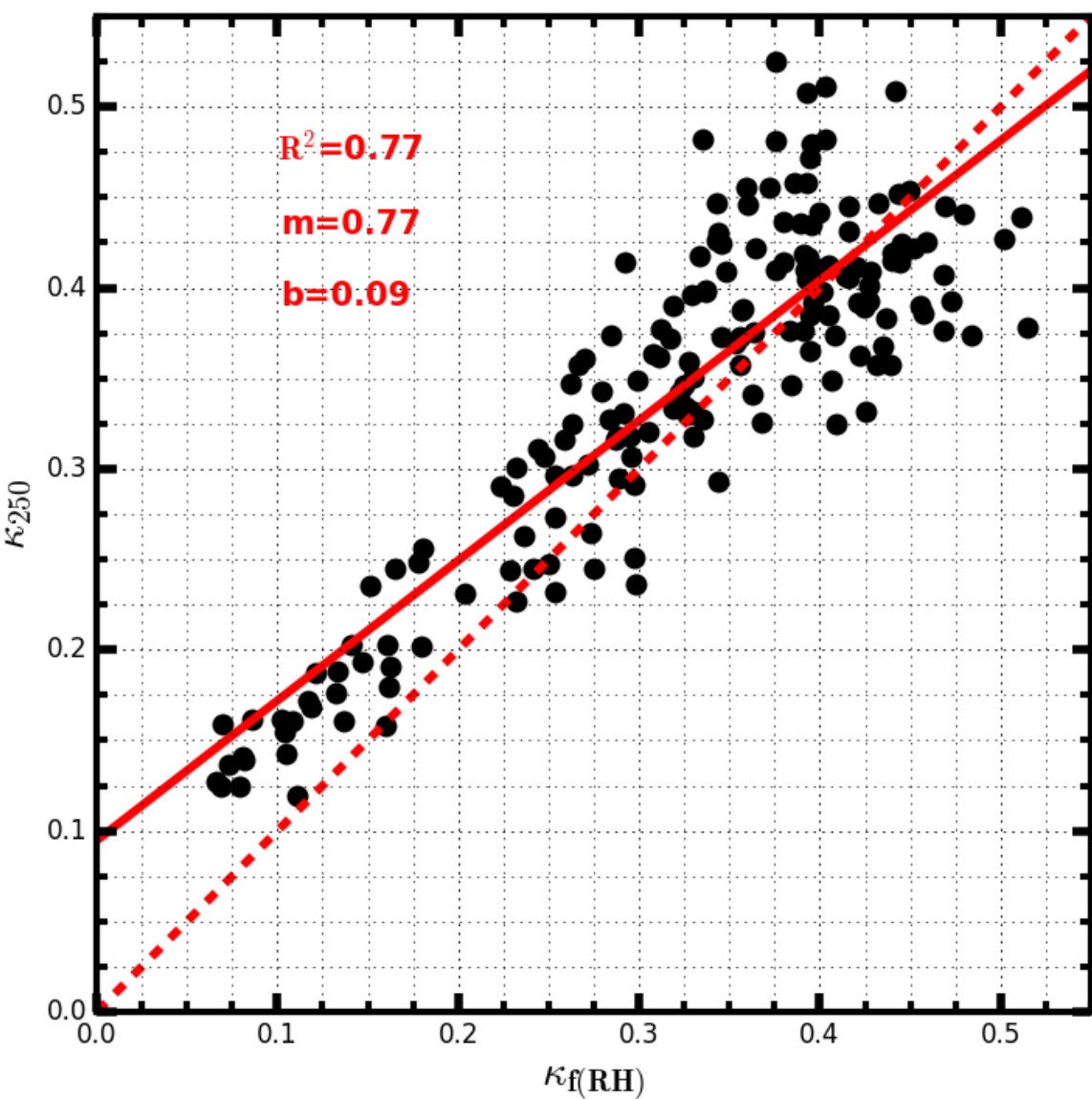


**Figure 2**. The comparison between $\kappa$ values derived from $f(\mathrm{RH})$ measurements ($\kappa_{f(\mathrm{RH})}$) and average $\kappa$ values for aerosol particles with a diameter of 250 nm ($\kappa_{250}$) which are derived from measurements of HH-TDMA. $R^2$ is the square of correlation coefficient, m is the slope and b is the intercept.

605

606

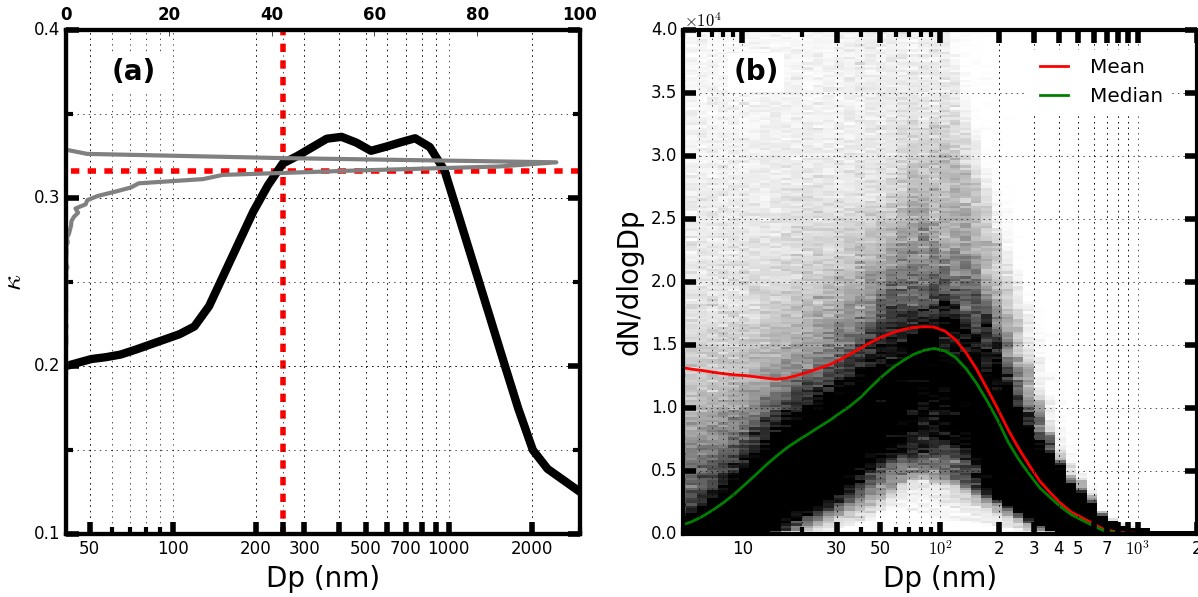

Figure 3. (a) The thick black line represents the average size-resolved $\kappa$ distribution from HaChi campaign. The solid gray line represents the probability distribution of retrieved $\kappa$ values with this size-resolve $\kappa$ distribution by using all PNSDs shown in figure (b), and the horizontal dashed line represents their average. The vertical dashed red line represents the position of 250 nm. (b) All PNSDs which are observed from three different representative background sites of the NCP during summer, they are used to model relationship between size-resolved $\kappa$ and retrieved $\kappa$ values from $f(\text{RH})$ measurements, and the gray color represents the frequency of PNSD, darker point corresponds to higher frequency, red and green line represent mean of median values of all observed PNSDs.

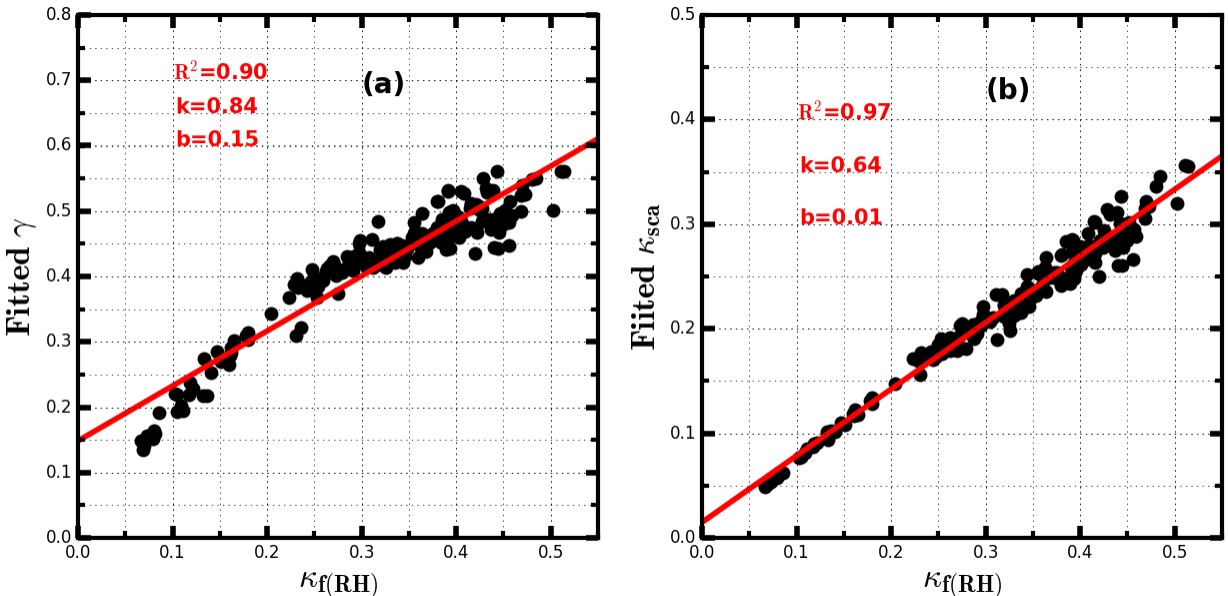

Figure 4. (a) The linear relationship between values of $\kappa_{f(\text{RH})}$ and fitted $\gamma$, $R^2$ is the square of correlation coefficient, k is the slope and b is the intercept; (b) The linear relationship between values of $\kappa_{f(\text{RH})}$ and fitted $\kappa_{sca}$.

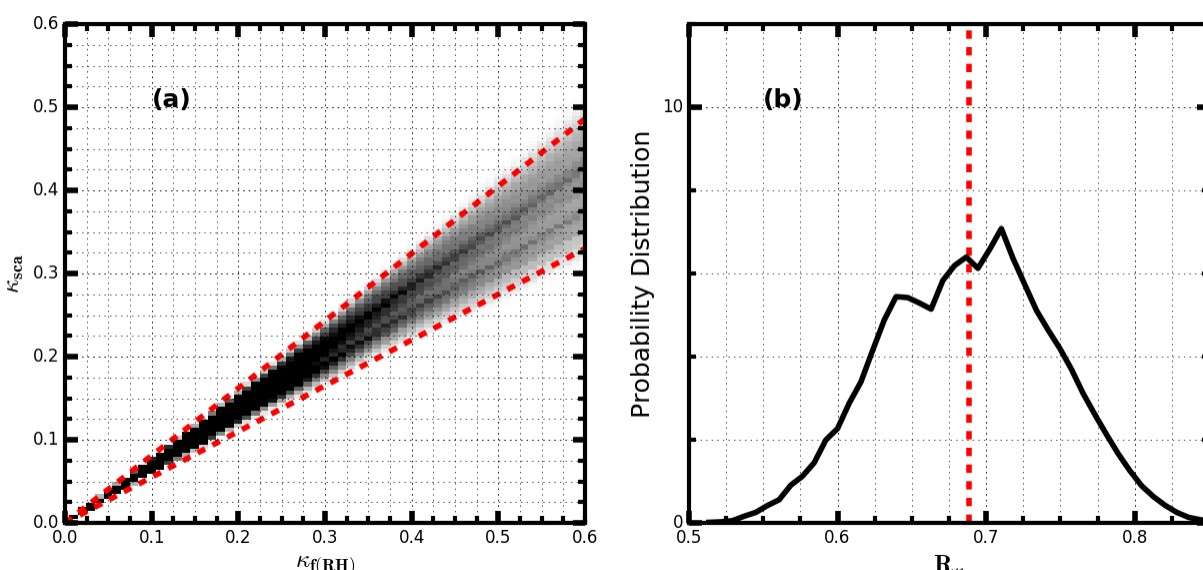

626

**Figure 5**. (a) Simulated relationships between $\kappa_{f(\text{RH})}$ and $\kappa_{sca}$ under different PNSD conditions ( all PNSDs shown in Fig.3a are used as inputs to conduct the simulation experiment), gray color represents the frequency and darker point corresponds to higher frequency, the slope of two dashed lines are 0.55 and 0.81; (b) The probability distribution of $R_\kappa$ ($\kappa_{sca}/\kappa_{f(\text{RH})}$).

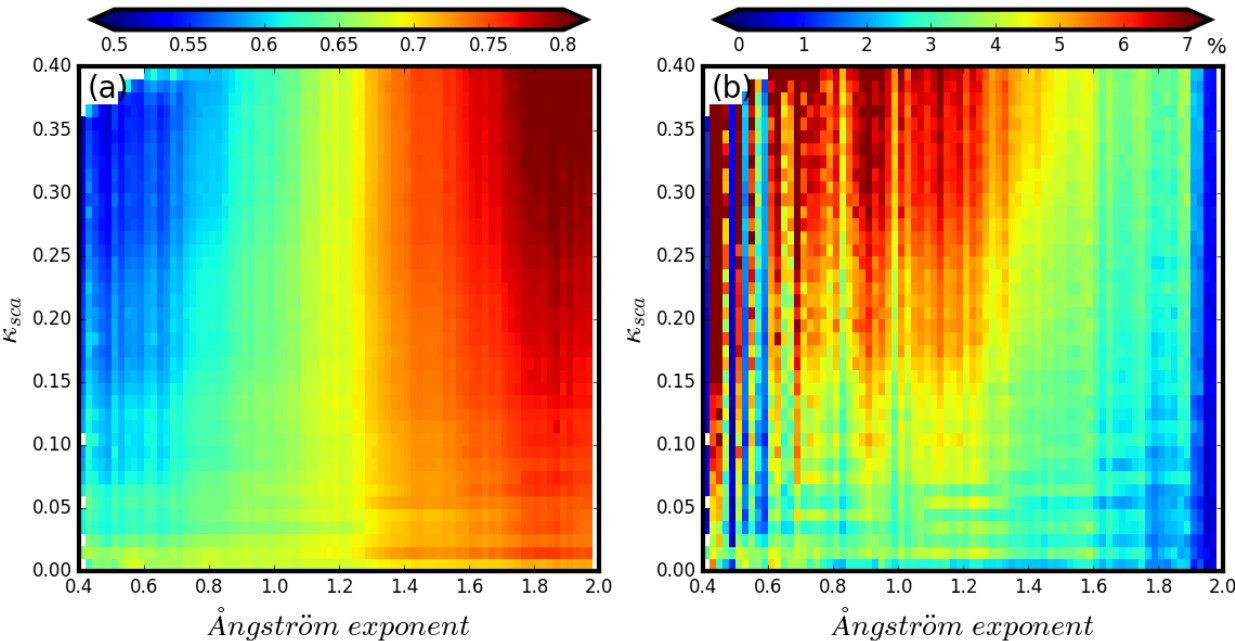

642

**Figure 6**. (a) Colors represent $R_\kappa$ values and the color bar is shown on the top of this figure, x-axis represents Ångström exponent and y-axis represents $\kappa_{sca}$. (b) Meanings of x-axis and y-axis are same with them in (a), however, color represents the percentile value of the standard deviation of $R_\kappa$ values within each grid divided by their average.

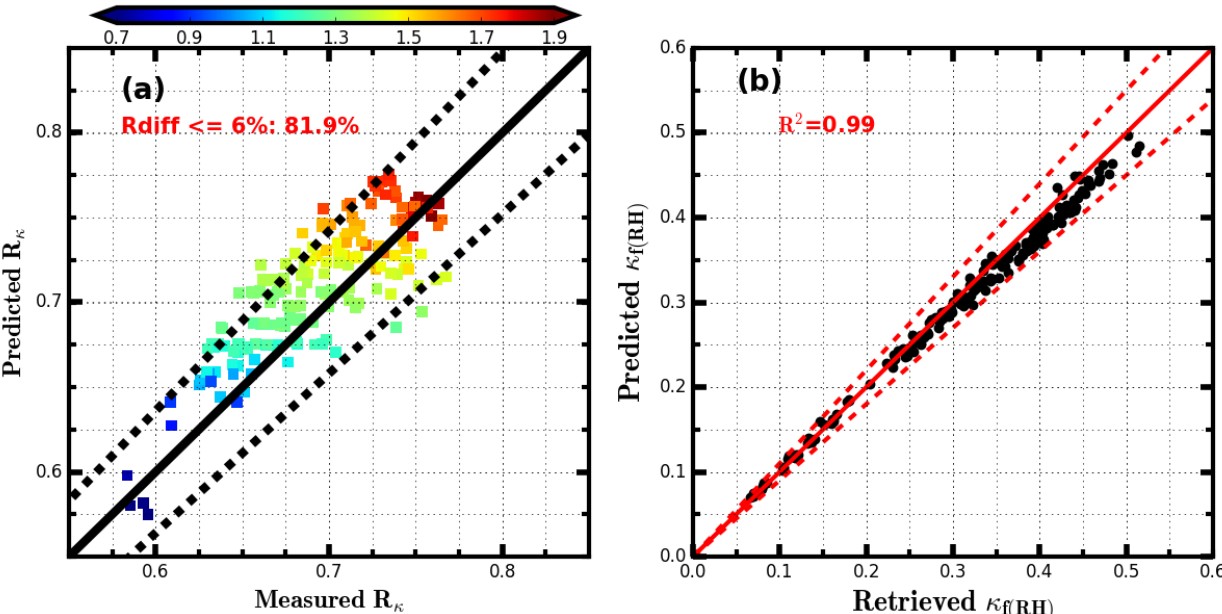

649

**Figure 7**. (a) The comparison between measured and predicted $R_\kappa$ values, colors represent values of Ångström exponent, texts with red color show the percentile of points with relative difference (Rdiff) less than 6% , two dashed line are 6% relative difference lines ; (b) the comparison between $\kappa_{f(RH)}$ retrieved from Method 1 and predicted $\kappa_{f(RH)}$ by using the new method introduced in Sect.4.3, $R^2$ is the square of correlation coefficient, two dashed lines are 10% relative difference lines.

655

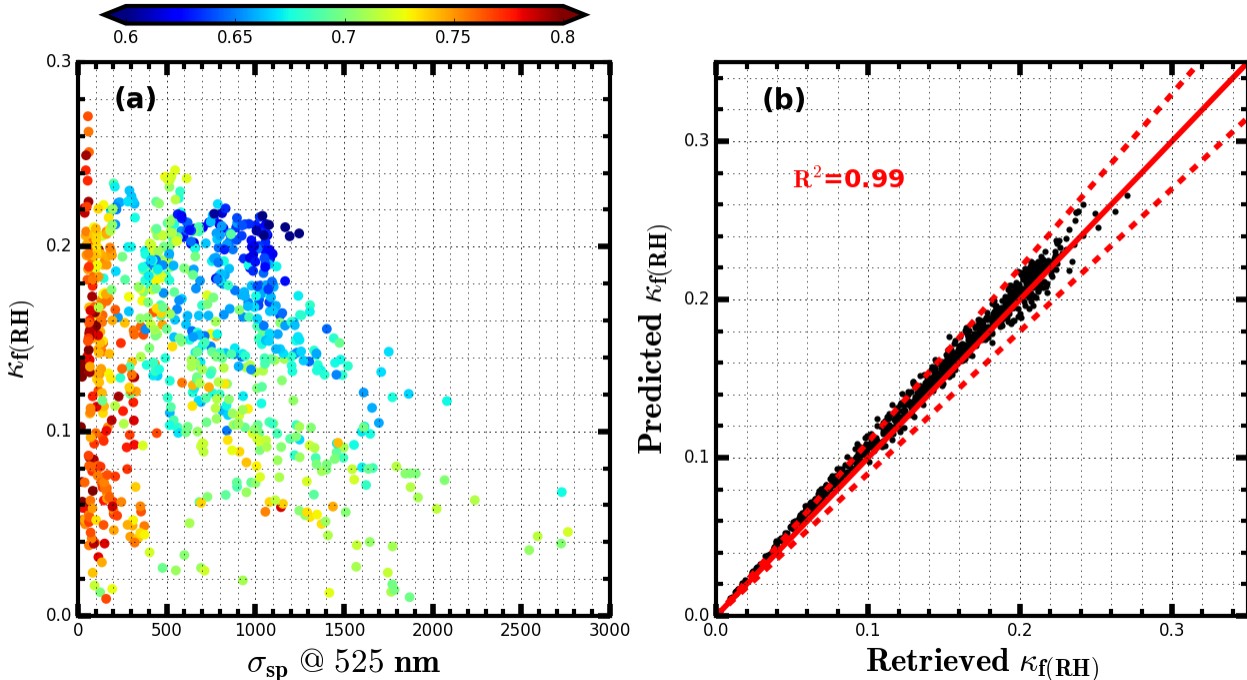

656

Figure 8. (a) x axis represents $\sigma_{sp}$ at 525 nm ($Mm^{-1}$), y axis represents retrieved $\kappa_{f(\text{RH})}$, colors of scatter points represent corresponding values of $R_{\kappa}$; (b) The comparison between $\kappa_{f(\text{RH})}$ retrieved from Method 1 and predicted $\kappa_{f(\text{RH})}$ by using the new method, $R^2$ is the square of correlation coefficient, two dashed red lines represent 10% deviations from 1:1 line.