# Peer review of "A novel method to derive the aerosol hygroscopicity parameter based only on"

_Atmospheric Chemistry and Physics, 2016_

## Referee Comment (RC1) · Anonymous Referee #3 · 17 Feb 2017

*Review:* " A novel method to derive the aerosol hygroscopicity parameter based only on measurements from a humidified nephelometer system"
*Authors:* Y. Kuang et al.

The authors present 2 methods for calculating the κ aerosol hygroscopicity parameter: from 1) Mie calculations that use the aerosol dry size distribution, BC mass concentration and the aerosol scattering hygroscopic growth (*fRH*) and 2) aerosol size-dependent hygroscopic growth, *gRH*. The authors use an empirical relationship between the *fRH* fit parameter, the scattering Angstrom exponent and the ratio fit values from *fRH* : method 1 kappa fit to create a look up table to predict kappa values.

The paper needs revision to better organize the paper, clarify the different hygroscopic fit calculations as well as discuss differences between the three kappa values.

In general: Break down run-on sentences into two or more sentences. Remove irrelevant information and words. Try not to repeat information. Reduce use of expressions such as "although, therefore, however, widely, especially and traditionally" as these terms usually don't carry meaning and make the paper more difficult to read. Try to use precise words rather than generalities.

Comments by section

Introduction

Lines 38-42: rewrite as "…,water usually constitutes half of the aerosol mass at a relative humidity of 80% with substantially higher water mass fractions existing at RH values above 90% for most ambient aerosol (Bian et al.,2014). The water content of aerosol and cloud droplets depends on both the ambient RH and hygroscopicity of the aerosol chemical constituents."

Line 45: rewrite as " In order to account for the mixed organic and inorganic composition of ambient aerosol Petters and Kriedweiss (207) proposed a modified version of Kohler theory called κ-Kohler theory to describe a single aerosol hygroscopic growth parameter, κ. The κ-Kohler equation, expressed in terms of the diameter growth factor, *g(RH),* is given in equation 1 below."

Equation 1: Please change "S" to RH/100. "S" is associated with droplet activation and may confuse readers. Remove "g" from the equation as it doesn't belong.

Line 58: remove sentence "In recent ten years, this…." as it states the obvious and doesn't add to the paper.

Line 71: rewrite as " The Humidified Tandem Differential Mobility Analyzer (HTDMA) measures the aerosol diameter hygroscopic growth as a function of RH.

Page 3: Remove reference to CCN measurements as it adds confusion and detracts from the discussion of diameter hygroscopic growth.

Line 84: Remove the Brock et al. reference as he uses a cavity ring down spectrometer and not a nephelometer.

Line 87-89: Rewrite as "The scattering enhancement factor $f(RH)$, defined as $f(RH) = \sigma sp(RH, \lambda)/ = \sigma sp (dry, \lambda)$, characterizes changes in the aerosol scattering coefficient with RH.

Line 92: Break into 2 sentences and rewrite as "Thus, $\kappa$ calculated from $f(RH)$ measurements represents an optically weighted aerosol hygroscopic growth.

Line 95-99: Don't start a new paragraph. Rewrite as "Traditionally, derivation of $\kappa$ from $f(RH)$ measurements requires aerosol PNSD as well as black carbon (BC) measurements to determine the imaginary part of the refractive index. As PNSD and BC measurements are expensive their availability in field campaigns are limited.

Clearly identify the 3 methods you use to determine $\kappa$ by identifying them as Method 1, Method 2 and Method 3.

Line 99: Start a new paragraph. "In this paper we use measurements from …. to derive $\kappa$ values using 3 methods. The first 2 methods derive $\kappa$ from aerosol diameter hygroscopic growth and the third method derives an aerosol optical parameterization of $\kappa$. Method 1, labeled as $\kappa fRH$, derives $\kappa$ from aerosol PNSD, BC and nephelometer $f(RH)$ measurements. Method 2, defined as $\kappa 250$, derives $\kappa$ from aerosol diameter hygroscopic growth measurements, $g(RH)$, using a High-Humidity Differential Mobility Analyzer (HH-TDMA). Method 3, defined as $\kappa scat$, is an empirical determination of $\kappa$ using only nepehelometer measurements of the aerosol scattering coefficient as a function of RH"

Start a new paragraph to describe how you combine $\kappa$ values from Methods 1 and 3 to devise a method to predict size-related $\kappa$ values using only $f(RH)$ scattering measurements. You need to clearly identify and separate these 3 methods in the paper. Using the terms "Method 1, Method 2 and Method 3" or something similar will clarify and simplify much of the paper discussion.

Page 5:
You need to describe the nephelometer $fRH$ measurements. What was the RH range and did the instruments operated in parallel or in series? Describe the position of the RH sensors, the type of RH sensor and its uncertainty. How was the RH inside the nephelometer determined? Did the humidifier scan the hydration or dehydration branch of the aerosol RH growth? What range of RH values were used in calculating the fits?

Page 6: Methodology

Why are the HTDMA measurements done at such a high RH? At RH values >90%, most RH sensors have an uncertainty +/- 3% or more. At high RH values the aerosol growth curve is particularly steep such that even a small error in RH would lead to a very high

error in gRH. What not measure gRH at a lower RH such as 80-85%? What is the uncertainty in gRH at 98% ?

Line 172: Remove the first two sentences of section 3.2. Accurate measurement of *fRH* depends on the uncertainty in aerosol scattering and RH. The empirical relationship of scattering to RH isn't difficult to measure or describe. What's difficult is modeling the size-dependent chemical composition of the aerosol, not the measurement itself.
Page 7, line 187: Remove the sentence " Here, we give …" As you haven't described curvature effects and these effects aren't apparent for equation 3, you should remove the sentence.
Line 189-198: Simplify the wording.

One assumption of the *fRH* kappa parameterization is that *fRH*=1 at RH=0. However, *fRH* values are near constant for RH <40%, meaning *fRH*=1 at RH $\sim$ 40. What this means is that the fits can't be forced through 1 at RH=0. The actual equation should be *fRH* = b + κ(RH/100-RH). Equation 3 doesn't account for aerosol losses in the humidifier and nephelometer. These losses won't affect the gamma fit parameter in equation 2 (provided losses are a percent of the scattering), but will affect determination of κ in equation 3. For example a 10% aerosol loss will change κ by 10%, but multiplying equation 2 by this same 10% loss correction or 1.1 won't change gamma.

The gamma and kappa fit parameters are sensitive to the RH range of the fit. What range of RH was used in the fits? Note that RH values <40% and >90% will increase error in the fit as the growth curves don't conform to equations 2 or 3 in these RH regions.

Line 225-226: rewrite: During deliquescence *f(RH)* exhibits an abrupt increase between RH values of 60-65%. As such, only *f(RH)* data points with RH >70% were used in determination of κ when deliquescence was apparent.

Lines 237-240. Be more specific and describe the hygroscopic growth behavior during polluted times more quantitatively. What range of σsp values were categorized as polluted? Figure 1 shows that σsp was above 100 Mm-1 most of the measurement period. Can you show a plot of frh vs σsp? Can you account for changes in *fRH* with aerosol loading? How does aerosol size and absorption change with loading?

Line 271: Is κ$_{fRH}$ optically weighted or is it a size-dependent κ that is integrated over the entire size distribution? Method 1 varies the size-dependent κ until the Mie calculations equal the scattering *fRH*. Change"optically weighted" to "size-integrated".

Page 10,Lines 266-285: Can you simplify the wording to make this paragraph easier to understand. It would help to distinguish the model κ values from Model 1and Model 2 if you labeled it κ*chem.*

Page 11, Line 299: A comparison of the kappa and gamma fits to the measured value at a single RH of 85%  isn't a good indication of the goodness of fit. Note in Figure 4a that both the gamma and kappa fits are higher than the measured value at 85%. A better

indication of the goodness of fit would be a chi-square fit value or the sum of the square of standard deviation of the measured values from the fit line or variance. I suggest replacing Figure 4b with a plot of the probability distribution of the fit variances.

Lines 313-321: This paragraph is unclear. The co-variance of *f(RH) fit* parameters with OMF, SMF and NMF varies with aerosol type; e.g. source and oxidation state or aging. The fit quality depends on the measurement duration as well as the variability in the aerosol type. The chemical information can give an indication of the aerosol hygroscopic growth in the absence of scattering *f(RH)* measurements. Gamma and kappa fit values in your comparison are both derived from nephelometer scattering measurements, so they should compare well. Remove the discussion of past comparisons of kappa with aerosol chemical composition.

Line 338: The ratio $R_k$ depends on the aerosol size-integrated scattering efficiency. This ratio will vary substantially with the aerosol type and size distribution. The variability of $R_k$ of this study may not coincide with that of the Brock et al. paper as different aerosol types were sampled under very different conditions. Rewrite sentence as " … the ratio κ-*scat/ κf(RH)* may share a similar range of variability."

Page 13, line 379: replace "PNSD at dry state" with "dry scattering Angstrom exponent" Line 380: remove "nevertheless, aerosol hygroscopicty has non-negligible impacts". The results indicate a strong size-dependence to the hygroscopic growth. Size and hygroscopicty are not separable, nor does one parameter dominate variation in $R_k$. Aerosol size would determine or dominate $R_k$ only if the aerosol chemical composition had no size variation. As sulfate tends to be more prevalent in smaller sizes then $R_k$ will be larger at higher Angstrom values.

Lines 378-384: Rewrite this section in terms of variation in the size-dependent chemical composition.

Lines 407-413: Note that the look up table only applies to aerosol on the NCP during the summer and can't be applied to other sites. Aerosol size distributions, secondary processing, and size-dependent composition vary widely with season and region. This method can be used as a tool for other sites, however it requires measurements of nephelometer scattering, aerosol BC and particle number size distributions.

---

## Referee Comment (RC2) · Anonymous Referee #2 · 9 Mar 2017

In this paper the authors present a novel method to derive the aerosol hygroscopicity parameter based on measurements from a nephelometer tandem. The topic is of interest for the scientific community; however, the performance of the proposed method is not sufficiently addressed and its general validity under different atmospheric conditions and predominant aerosol types is not clear. The paper needs major revision to improve scientific aspects of the presented method but also to improve the organization, grammar and readability of the manuscript.

General comment: If the focus of the manuscript, as stated in the title, is the presentation of a novel method they need to pay more attention to the explanation of the method itself and the validation of the method using additional data (ideally from different mea-

surement sites) and quantify the uncertainties of using the look-up-table to retrieve aerosol hygroscopicity. Otherwise, the authors are just presenting relationships between variables but not actually a new (usable) method. There are many redundancies in the paper that should be omitted as well as typos and grammar spelling errors. Split sentences into two or more individual sentences to improve readability.

Specific comments: Line21: avoid redundancies like "newly proposed novel approach"

Line22: Replace by "... is that $\kappa$f(RH) can be estimated without any additional information..."

Line34: "...most important factors affecting these..." Introduction: there are too much methodological information in the introduction, that should be moved to the methodology section.

Line104: "similar to"

Line107: "based on"

Section2: This section is very bad organized. Include a table with information about the campaigns (dates, sites, data used here from each campaign, etc). What is the time resolution of the PM2.5 filter samples? 24 hours? How often is the sampling performed?

More information on HH-TDMA measurements and inversion routine should be presented. Same applies for the nephelometers tandem. Include information on nephelometers correction and calibration, humidogram schedule, RH range in the dry neph, where were the RH sensors ocated in the system?, how often were the sensors calibrated?

Line120: replace dot by comma

Section 3.1: Further details on the methods used to derive the $\kappa$ parameter should be given even though the methods were published before. At least the basic information to

allow the reader to understand the manuscript. Concerning the $\kappa$f(RH) method, which chemical species have been considered apart of BC? A table including the chemical species, refractive indices, densities and contribution during the measurement period must be included.

In the Mie routine, is the chemistry considered as constant during the campaign? See my previous comment on PM2.5 sampling schedule.

Section 3.2: The reference of Quinn et al. (2005) is not appropriate here. The gamma parameterization was first introduced in Kasten (1969) and Hanel (1980). Kasten, F., 1969. Visibility forecast in the phase of pre-condensation. Tellus 21 (5), 631-635 Hanel, G., 1980. Technical Note: an attempt to interpret the humidity dependencies of the aerosol extinction and scattering coefficients. Atmos. Environ. 15, 403-406

Line 194: avoid redundancy, this sentence "more details. . ." could be omitted.

Results: Line207-221: This paragraph could be omitted since basically is a repetition of the results presented in Kuang et al., (2016) and does not provide any additional/useful information.

Line 207: information about nephelometer correction should be moved to the instrument section.

Lines 212, 216 and somewhere else: "a lot" is not very scientific, be more quantitative and avoid colloquial expressions.

Line 257: This paragraph should be rewritten. What is the aim of including these two additional campaigns?

Line 297: "The fitting performance. . . values" could be omitted. Again, avoid redundancy.

Line 300: The $\gamma$-Method and $\kappa$-Method are just different ways of fitting the experimental f(RH)-RH relationship. Which method is better or worst depends on your specific data,

and many other equations have been previously proposed in the literature (Titos et al., 2016). The discussion in lines 300-306 and figure 4 about which fitting is best do not add much and could be omitted.

Line 316: "pretty good linear relationship" does not sound very quantitative neither scientific. . .. Try to be more specific. . .

Line 333: This is the first time that $\kappa$chem and $\kappa$ext are introduced.

Line 347 and somewhere else: Avoid repetitions like "which is introduced in Section . . ."

Line 359-360: "and then it turns out", "much more complex". . . this is not very appropriate for a scientific paper. . .

References of Titos et al., 2016 and Zieger et al., 2014 are not used appropriately here. The Angstrom exponent was first introduced by Angstrom!

Line 377: Keep in mind that the Angstrom exponent is not a measure of the PNSD, it provides information on mean predominant aerosol size so values close to 2 denote a predominance of fine particles while values below 1 denote a predominance of coarse particles.

Line 393 and Figure 7: This comparison exercise is interesting but it is not appropriately done. The predicted Rk values using the look-up-table are compared with the measured Rk values. However, these measured Rk were used before to generate the look-up-table. Thus, it is clear that a high correlation is expected. A different dataset, with additional Rk values not used to generate the look-up-table should be used for validation of the proposed model. Otherwise, the same data that is used to generate the model is used to validate it, which is meaningless.

If the authors really expect researchers to use their method, they should provide them with an uncertainty range for Rk as a function of the Angstrom exponent and $\kappa$sca. Probably, higher errors are expected at higher $\kappa$sca values? This is certainly needed

if they expect people to use the look-up-table. In general, the manuscript lacks of an appropriate treatment of errors despite the large expected errors for the hygroscopicity parameters.
* * *

---

## Author Comment (AC1) · 11 Apr 2017

The responses, the revised manuscript and the supporting information are included in the supplement zip file.

Please also note the supplement to this comment: http://www.atmos-chem-phys-discuss.net/acp-2016-1066/acp-2016-1066-AC1-supplement.zip

---

## Author Response (AR1)

**Response to anonymous referee #3**

**Main comments**: The authors present 2 methods for calculating the κ aerosol hygroscopicity parameter: from 1) Mie calculations that use the aerosol dry size distribution, BC mass concentration and the aerosol scattering hygroscopic growth (fRH) and 2) aerosol size-dependent hygroscopic growth, gRH. The authors use an empirical relationship between the fRH fit parameter, the scattering Angstrom exponent and the ratio fit values from fRH : method 1 kappa fit to create a look up table to predict kappa values.

The paper needs revision to better organize the paper, clarify the different hygroscopic fit calculations as well as discuss differences between the three kappa values.

In general: Break down run-on sentences into two or more sentences. Remove irrelevant information and words. Try not to repeat information. Reduce use of expressions such as "although, therefore, however, widely, especially and traditionally" as these terms usually don't carry meaning and make the paper more difficult to read. Try to use precise words rather than generalities.

**Response**: Thanks for your comment. We have revised the manuscript according to your suggestions.

**Specific comments**

**Comment**: Lines 38-42: rewrite as "…,water usually constitutes half of the aerosol mass at a relative humidity of 80% with substantially higher water mass fractions existing at RH values above 90% for most ambient aerosol (Bian et al.,2014). The water content of aerosol and cloud droplets depends on both the ambient RH and hygroscopicity of the aerosol chemical constituents."

**Response**: Thanks for your comment. We have revised the manuscript accordingly.

**Comment**: rewrite as " In order to account for the mixed organic and inorganic composition of ambient aerosol Petters and Kriedweiss (207) proposed a modified version of Kohler theory called κ-Kohler theory to describe a single aerosol hygroscopic growth parameter, κ. The κ-Kohler equation, expressed in terms of the diameter growth factor, g(RH), is given in equation 1 below."

**Response**: Thanks for your suggestion. We revised the manuscript accordingly.

**Comment**: Equation 1: Please change "S" to RH/100. "S" is associated with droplet activation and may confuse readers. Remove "g" from the equation as it doesn't belong

**Response**: Thanks for your comment. We revised the manuscript accordingly.

**Comment**: Line 58: remove sentence "In recent ten years, this…." as it states the obvious and doesn't add to the paper.

**Response**: Thanks for your suggestion. We revised the manuscript accordingly.

**Comment**: Line 71: rewrite as " The Humidified Tandem Differential Mobility Analyzer (HTDMA) measures the aerosol diameter hygroscopic growth as a function of RH.

**Response**: Thanks for your suggestion. We revised the manuscript accordingly.

**Comment**: Page 3: Remove reference to CCN measurements as it adds confusion and detracts from the discussion of diameter hygroscopic growth.

**Response**: Thanks for your comment. We revised the manuscript accordingly.

**Comment**: Line 84: Remove the Brock et al. reference as he uses a cavity ring down spectrometer and not a nephelometer.

**Response**: Thanks for your comment. We revised the manuscript accordingly.

**Comment**: Line 87-89: Rewrite as "The scattering enhancement factor f(RH), defined as $f(RH) = \sigma_{sp}(RH, \lambda)/\sigma_{sp}(dry, \lambda)$, characterizes changes in the aerosol scattering coefficient with RH.

**Response**: Thanks for your comment. We revised the manuscript accordingly.

**Comment**: Line 92: Break into 2 sentences and rewrite as "Thus, κ calculated from f(RH) measurements represents an optically weighted aerosol hygroscopic growth.

**Response**: Thanks for your comment. We revised the manuscript accordingly.

**Comment**: Line 95-99: Don't start a new paragraph. Rewrite as "Traditionally, derivation of κ from f(RH) measurements requires aerosol PNSD as well as black carbon (BC) measurements to determine the imaginary part of the refractive index. As PNSD and BC measurements are expensive their availability in field campaigns are limited.

**Response**: Thanks for your suggestion. We revised the manuscript accordingly.

**Comment**: Clearly identify the 3 methods you use to determine κ by identifying them as Method 1, Method 2 and Method 3. Line 99: Start a new paragraph. "In this paper we use measurements from …. to derive κ values using 3 methods. The first 2 methods derive κ from aerosol diameter hygroscopic growth and the third method derives an aerosol optical parameterization of κ. Method 1, labeled as $\kappa_{f(RH)}$, derives κ from aerosol PNSD, BC and nephelometer f(RH) measurements. Method 2, defined as κ250, derives κ from aerosol diameter hygroscopic growth measurements, g(RH), using a High-Humidity Differential Mobility Analyzer (HH-TDMA). Method 3, defined as $\kappa_{sca}$, is an empirical determination of κ using only nepehelometer measurements of the aerosol scattering coefficient as a function of RH".

Start a new paragraph to describe how you combine κ values from Methods 1 and 3 to devise a method to predict size-related κ values using only f(RH) scattering measurements. You need to clearly identify and separate these 3 methods in the paper. Using the terms "Method 1, Method 2 and Method 3" or something similar will clarify and simplify much of the paper discussion.

**Response**: Thanks for your comment. We revised the manuscript accordingly.

**Comment**: You need to describe the nephelometer fRH measurements. What was the RH range and did the instruments operated in parallel or in series? Describe the position of the RH sensors, the type of RH sensor and its uncertainty. How was the RH inside the nephelometer determined? Did the humidifier scan the hydration or dehydration branch of the aerosol RH growth? What range of RH values were used in calculating the fits?

**Response**: Thanks for your comment. We have added these information in the Section 2 of the revised manuscript.

**Comment**: Why are the HTDMA measurements done at such a high RH? At RH values >90%, most RH sensors have an uncertainty +/- 3% or more. At high RH values the aerosol growth curve is particularly steep such that even a small error in RH would lead to a very high error in g(RH). What not measure g(RH) at a lower RH such as 80-85%? What is the uncertainty in g(RH) at 98% ?

**Response**: Thanks for your comment. The basic principle of HH-TDMA is similar to that of HTDMA, however, its special feature is capable of operating stably under extremely high RH conditions (Hennig et al., 2005). The reason that this system operates at RH of 98% is the scientific focus of this instrument during this field campaign is hygroscopic properties of aerosol particles under extremely high RH conditions. Details about the uncertainties of RH and g(RH) please refer to Hennig et al. (2005).

**Comment**: Line 172: Remove the first two sentences of section 3.2. Accurate measurement of f(RH) depends on the uncertainty in aerosol scattering and RH. The empirical relationship of scattering to RH isn't difficult to measure or describe. What's difficult is modeling the size-dependent chemical composition of the aerosol, not the measurement itself. Page 7, line 187: Remove the sentence " Here, we give …" As you haven't described curvature effects and these effects aren't apparent for equation 3, you should remove the sentence.

**Response**: Thanks for your comment. We have revise the manuscript accordingly.

**Comment**: Line 189-198: Simplify the wording.

**Response**: Thanks for your comment. We have revised the manuscript.

**Comment**: One assumption of the f(RH) kappa parameterization is that fRH=1 at RH=0. However, f(RH) values are near constant for RH <40%, meaning f(RH)=1 at RH ~ 40. What this means is that the fits can't be forced through 1 at RH=0. The actual equation should be f(RH) = b + κ(RH/100-RH). Equation 3 doesn't account for aerosol losses in the humidifier and nephelometer. These losses won't affect the gamma fit parameter in equation 2 (provided losses are a percent of the scattering), but will affect determination of κ in equation 3. For example a 10% aerosol loss will change κ by 10%, but multiplying equation 2 by this same 10% loss correction or 1.1 won't change gamma.

**Response**: Thanks for your comment. We agree with the referee that the humidified nephelometer system have the problem of aerosol losses in the humidifier. During Gucheng campaign (introduced in the revised manuscript), the control software of this system will let the humidifier do not humidify the sample air every two days and the period last about two hours. The purpose of doing so is to check the consistency of two nephelometers (Dry nephelometer and Wet Nephelometer). The results are shown

[Figure]

**Figure 1**. x-axis represents $\sigma_{sp}$ at 525 nm measured by the dry nephelometer, y-axis represents $\sigma_{sp}$ at 525 nm measured by the wet nephelometer, the red line is 1:1 line.

in Fig.1. The results demonstrate that $\sigma_{sp}$ measured by the wet nephelometer is slightly higher than that measured by the dry nephelometer, the average relative difference is 3%. The reason that the higher $\sigma_{sp}$ measured by the wet nephelometer might be attributed to the difference of RH in the dry nephelometer (about 8%) and wet nephelometer (about 15%). In addition, the relative difference between them is within the measurement uncertainty of nephelometer (Müller et al., 2011). This result indicates that aerosol losses in the humidifier have negligible influence on the $\sigma_{sp}$.

During processes of measuring $f(\text{RH})$, the sample RH in the dry nephelometer ($RH_0$) is not zero. We have modified the fitting formula of measured $f(\text{RH})$. According to equation (3) of the manuscript, the measured $f(\text{RH})_{measure} = \frac{f(\text{RH})}{f(RH_0)}$ should be fitted using the following formula:

$$f(\text{RH})_{measure} = (1 + \kappa_{sca}\frac{RH}{100-RH})\big/(1 + \kappa_{sca}\frac{RH_0}{100-RH_0}) \quad (4)$$

And in the revised manuscript, this equation is used for calculating $\kappa_{sca}$.

**Comment**: The gamma and kappa fit parameters are sensitive to the RH range of the fit. What range of RH was used in the fits? Note that RH values <40% and >90% will increase error in the fit as the growth curves don't conform to equations 2 or 3 in these RH regions.

**Response**: Thanks for your comment. We have added the information about RH range used in the retrieval algorithm in the revised manuscript. About 50% to 90% for cycles without deliquescence, about 70% to 90% for cycles with deliquescence.

**Comment**: Line 225-226: rewrite: During deliquescence f(RH) exhibits an abrupt increase between RH values of 60-65%. As such, only f(RH) data points with RH >70% were used in determination of κ when deliquescence was apparent.

**Response**: Thanks for your comment. We revised the manuscript accordingly.

**Comment**: Lines 237-240. Be more specific and describe the hygroscopic growth behavior during polluted times more quantitatively. What range of $\sigma_{sp}$ values were categorized as polluted? Figure 1 shows that $\sigma_{sp}$ was above 100 Mm-1 most of the measurement period. Can you show a plot of f(RH) vs $\sigma_{sp}$? Can you account for changes in fRH with aerosol loading? How does aerosol size and absorption change with loading?

**Response**: Thanks for your comment. Periods with $\sigma_{sp}$>100 Mm-1 are categorized as polluted.

[Figure]

Figure 2. x-axis represents $\sigma_{sp}$ @ 550 nm, y-axis represents retrieved $\kappa_{f(RH)}$.

During Wangdu campaign, the plot of $\kappa_{f(RH)}$ vs $\sigma_{sp}$ is shown in Fig.2. The aerosol size and absorption changes with aerosol loading is a good scientific topic. Studies of aerosol size changes with aerosol loading can be found in previous studies, such as Shen et al. (2015). The absorption change with aerosol loading is highly variable in polluted region due to complicated emissions and aging processes. Issues about aerosol size and absorption changes with loading are beyond the scope of this paper.

**Comment**: Line 271: Is $\kappa_{f(RH)}$ optically weighted or is it a size-dependent $\kappa$ that is integrated over the entire size distribution? Method 1 varies the size-dependent $\kappa$ until the Mie calculations equal the scattering f(RH). Change "optically weighted" to "size-integrated".
**Response**: Thanks for your comment. We revised the manuscript accordingly.

**Comment**: Page 10,Lines 266-285: Can you simplify the wording to make this paragraph easier to understand. It would help to distinguish the model $\kappa$ values from Model 1and Model 2 if you labeled it $\kappa_{chem}$
**Response**: Thanks for your comment. We have revised the manuscript.

**Comment**: Page 11, Line 299: A comparison of the kappa and gamma fits to the measured value at a single RH of 85% isn't a good indication of the goodness of fit. Note in Figure 4a that both the gamma and kappa fits are higher than the measured value at 85%. A better indication of the goodness of fit would be a chi-square fit value or the sum of the square of standard deviation of the measured values from the fit line or variance. I suggest replacing Figure 4b with a plot of the probability distribution of the fit variances.
**Response**: Thanks for tour comment. We agree with the referee. According to the suggestions of another referee, we have deleted this part.

**Comment**: Lines 313-321: This paragraph is unclear. The co-variance of f(RH) fit parameters with OMF, SMF and NMF varies with aerosol type; e.g. source and oxidation state or aging. The fit quality depends on the measurement duration as well as the variability in the aerosol type. The chemical information can give an indication of the aerosol hygroscopic growth in the absence of scattering f(RH) measurements. Gamma and kappa fit values in your comparison are both derived from nephelometer scattering measurements, so they should compare well. Remove the discussion of past comparisons of kappa with aerosol chemical composition

**Response**: Thanks for your comment. We have revised the manuscript accordingly.

**Comment**: Line 338: The ratio $R_k$ depends on the aerosol size-integrated scattering efficiency. This ratio will vary substantially with the aerosol type and size distribution. The variability of Rk of this study may not coincide with that of the Brock et al. paper as different aerosol types were sampled under very different conditions. Rewrite sentence as " ... the ratio $\kappa_{sca}/\kappa_{f(RH)}$ may share a similar range of variability."

**Response**: Thanks for your suggestion. We revised the manuscript accordingly.

**Comment**: Page 13, line 379: replace "PNSD at dry state" with "dry scattering Angstrom exponent" Line 380: remove "nevertheless, aerosol hygroscopicty has non-negligible impacts". The results indicate a strong size-dependence to the hygroscopic growth. Size and hygroscopicty are not separable, nor does one parameter dominate variation in $R_k$. Aerosol size would determine or dominate $R_k$ only if the aerosol chemical composition had no size variation. As sulfate tends to be more prevalent in smaller sizes then $R_k$ will be larger at higher Angstrom values.

**Response**: Thanks for your suggestion. We have revised the manuscript accordingly.

**Comment**: Lines 378-384: Rewrite this section in terms of variation in the size-dependent chemical composition.

**Response**: Thanks for your comment. We agree with the referee that the size-dependent chemical composition also exerts influence on $R_k$. If PNSD is fixed, each size-resolved $\kappa$ distribution corresponds to a certain $\kappa_{f(RH)}$, and $\kappa_{f(RH)}$ varies within certain range no matter how size-resolved $\kappa$ distribution changes. Therefore, influences of size-dependent chemical compositions are already included in simulated results of producing the look up table by varying the $\kappa_{f(RH)}$ from 0 to 0.7 for a fixed aerosol PNSD. This discussion is added in the revised manuscript.

**Comment**: Lines 407-413: Note that the look up table only applies to aerosol on the NCP during the summer and can't be applied to other sites. Aerosol size distributions, secondary processing, and size-dependent composition vary widely with season and region. This method can be used as a tool for other sites, however it requires measurements of nephelometer scattering, aerosol BC and particle number size distributions.

**Response**: Thanks for your comment. We agree with the referee that aerosol size distributions, secondary processing, and size-dependent composition vary widely with season and region. However, in the simulating processes of producing the look up table, no information about size dependent chemical composition is involved. The look up shown in Fig.6a of the manuscript is produced from measurements of four field campaigns (datasets from a new campaign are added) which were conducted at different seasons and sites. The small variation of $R_k$ under different Angstrom exponent and $\kappa_{sca}$ conditions shown in Fig.6b demonstrate good consistency exists between $R_k$ produced from PNSD and BC measurements of these campaigns. In addition, in the revised manuscript, we have verified this method with datasets obtained from two sites of the NCP in different seasons. Please refer to Fig.7 and Fig.8 of the revised manuscript for more details. The results demonstrate that the look up table is applicable in different sites and seasons.

**Response to anonymous referee #2**

**General comment**: If the focus of the manuscript, as stated in the title, is the presentation of a novel method they need to pay more attention to the explanation of the method itself and the validation of the method using additional data (ideally from different measurement sites) and quantify the uncertainties of using the look-up-table to retrieve aerosol hygroscopicity. Otherwise, the authors are just presenting relationships between variables but not actually a new (usable) method. There are many redundancies in the paper that should be omitted as well as typos and grammar spelling errors. Split sentences into two or more individual sentences to improve readability.

**Response**: Thanks for you comment. We agree with the reviewer. In the revised manuscript, the look up table is produced based on PNSD and BC measurements from four different field campaigns. Meanwhile, datasets about PNSD and BC during Wangdu campaign when measurements from the humidified nephelometer system are available are not used in simulating the look up table. In addition, datasets from a different field campaign which is conducted at another site on the NCP in autumn is also used to validate the proposed method. That is, the produced look up table is verified with measurements from two different sites in different seasons. Please refer to Fig.7 and Fig.8 of the revised manuscript for more details. The results demonstrate that the look up table is applicable in different sites and seasons. As to uncertainties of $R_k$ under different Angstrom exponent and $\kappa_{sca}$ conditions, the standard deviations of $R_k$ within each grid of the look up table are shown in Fig.6b.

**Specific comments**

**Comment**: Line21: avoid redundancies like "newly proposed novel approach"

**Response**: Thanks for your suggestion. We revised the manuscript accordingly.

**Comment**: Line22: Replace by " . . . is that $\kappa_{f(RH)}$ can be estimated without any additional information…"

**Response**: Thanks for you suggestion. We revised the manuscript accordingly.

**Comment**: Line34: " . . .most important factors affecting these . . ." Introduction: there are too much methodological information in the introduction, that should be moved to the methodology section.

**Response**: Thanks for your suggestion. This sentence reflects the significance of aerosol hygroscopicity, and this is the motivation of this research.

**Comment**: Line104: "similar to"

**Response**: Thanks for you suggestion. We revised the manuscript accordingly.

**Comment**: Line107: "based on"

**Response**: Thanks for your comment. We revised the manuscript accordingly.

**Comment**: Section2: This section is very bad organized. Include a table with information about the campaigns (dates, sites, data used here from each campaign, etc). What is the time resolution of the PM2.5 filter samples? 24 hours? How often is the sampling performed?

**Response**: Thanks for your suggestion. We revised this section according to your suggestions.

**Comment**: More information on HH-TDMA measurements and inversion routine should be presented. Same applies for the nephelometers tandem. Include information on nephelometers correction and calibration, humidogram schedule, RH range in the dry neph, where were the RH sensors located in the system?, how often were the sensors calibrated?

**Response**: Thanks for your suggestion. We added information about the nephelometer system in Section 2 of the revised manuscript. The instrument set-up of HH-TDMA and inversion routine of $\kappa$ from measurements of HH-TDMA are introduced in detail in Liu et al. (2011).

**Comment**: Line120: replace dot by comma

**Response**: Thanks for your suggestion. We have revised the manuscript.

**Comment**: Section 3.1: Further details on the methods used to derive the $\kappa$ parameter should be given even though the methods were published before. At least the basic information to allow the reader to understand the manuscript. Concerning the $\kappa_{f(RH)}$ method, which chemical species have been considered apart of BC? A table including the chemical species, refractive indices, densities and contribution during the measurement period must be included.

**Response**: Thanks for your comment. The flow chart of retrieving $\kappa_{f(RH)}$ is provided in the supporting information. A simplified aerosol model was applied to aerosol optical calculations. In the model, aerosol components are divided into two classes in terms of their optical properties: the light absorbing component (BC) and less absorbing components (comprising inorganic salts and acids such as sulfates, nitrates, ammoniums, as well as most of the organic compounds). We have added this statement in Section 3.1 of the revised manuscript.

**Comment**: In the Mie routine, is the chemistry considered as constant during the campaign? See my previous comment on PM2.5 sampling schedule.

**Response**: Thanks for you comment. In this paper, a simplified aerosol model was applied to aerosol optical calculations. In the model, aerosol components are divided into two classes in terms of their optical properties: the light absorbing component (BC) and less absorbing components (comprising inorganic salts and acids such as sulfates, nitrates, ammoniums, as well as most of the organic compounds). We have added this statement in Section 3.1 of the revised manuscript.

**Comment**: Section 3.2: The reference of Quinn et al. (2005) is not appropriate here. The gamma parameterization was first introduced in Kasten (1969) and Hanel (1980). Kasten,F., 1969. Visibility forecast in the phase of pre-condensation. Tellus 21 (5), 631-635 Hanel, G., 1980. Technical Note: an attempt to interpret the humidity dependencies of the aerosol extinction and scattering coefficients. Atmos. Environ. 15, 403-406.

**Response**: Thanks for your comment. We have revised the reference accordingly.

**Comment**: Line 194: avoid redundancy, this sentence "more details . . ." could be omitted.

**Response**: Thanks for your comment. We have revised the manuscript accordingly.

**Comment**: Results: Line207-221: This paragraph could be omitted since basically is a repetition of the results presented in Kuang et al., (2016) and does not provide any additional/useful information.

**Response**: Thanks for your comment. We have deleted these sentences.

**Comment**: Line 207: information about nephelometer correction should be moved to the instrument section

**Response**: Thanks for your suggestion. We revised the manuscript accordingly.

**Comment**: Lines 212, 216 and somewhere else: "a lot" is not very scientific, be more quantitative and avoid colloquial expressions.

**Response**: Thanks for your comment. We have revised the manuscript accordingly.

**Comment**: Line 257: This paragraph should be rewritten. What is the aim of including these two additional campaigns?

**Response**: Thanks for your comment. The aim of including PNSD and BC information from different campaigns is to simulate variations of $R_\kappa$ under different conditions. We have added this sentence in the revised manuscript.

**Comment**: Line 297: "The fitting performance . . . values" could be omitted. Again, avoid redundancy.

**Response**:

**Comment**: Line 300: The $\gamma$-Method and $\kappa$-Method are just different ways of fitting the experimental f(RH)-RH relationship. Which method is better or worst depends on your specific data, and many other equations have been previously proposed in the literature (Titos et al.,2016). The discussion in lines 300-306 and figure 4 about which fitting is best do not add much and could be omitted.

**Response**: Thanks for your comment. We have deleted these sentences.

**Comment**: Line 316: "pretty good linear relationship" does not sound very quantitative neither scientific . . .. Try to be more specific …

**Response**: Thanks for your comment. We have revised the manuscript accordingly.

**Comment**: Line 333: This is the first time that $\kappa$chem and $\kappa$ext are introduced.

**Response**: Thanks for your comment. We revised the manuscript.

**Comment**: Line 347 and somewhere else: Avoid repetitions like "which is introduced in Section

. . .”

**Response**: Thanks for your comment. We have revised the manuscript accordingly.

**Comment**: Line 359-360: "and then it turns out", "much more complex"… this is not very appropriate for a scientific paper…

**Response**: Thanks for your comment. This sentences is revised as the following: "A robust linear relationship is found between $\kappa_{f(\mathrm{RH})}$ and $\kappa_{sca}$ in Sect.4.2 , however, results of further analysis suggest that $R_\kappa$ varies a lot"

**Comment**: References of Titos et al., 2016 and Zieger et al., 2014 are not used appropriately here. The Angstrom exponent was first introduced by Angstrom!

**Response**: Thanks for your comment. We have deleted the reference.

**Comment**: Line 377: Keep in mind that the Angstrom exponent is not a measure of the PNSD, it provides information on mean predominant aerosol size so values close to 2 denote a predominance of fine particles while values below 1 denote a predominance of coarse particles.

**Response**: Thanks for your comment. We have revised the sentence as the following: "Based on results shown on Fig.6a, the different impacts of aerosol hygroscopicity and dry scattering Ångström exponent on $R_\kappa$ can be distinguished to some extent".

**Comment**: Line 393 and Figure 7: This comparison exercise is interesting but it is not appropriately done. The predicted Rk values using the look-up-table are compared with the measured $R_k$ values. However, these measured Rk were used before to generate the look-up-table. Thus, it is clear that a high correlation is expected. A different dataset, with additional $R_k$ values not used to generate the look-up-table should be used for validation of the proposed model. Otherwise, the same data that is used to generate the model is used to validate it, which is meaningless.

**Response**: Thanks for your comment. In the revised manuscript. The dataset about PNSDs and mass concentrations of BC are not used in the processes of produing the look up table shown in Fig.6a. Thus, the look up table is independent of measurements during periods when f(RH) measurements are available. In addition, f(RH) measurements from another campaign is also used to verify the manuscript.

**Comment**: If the authors really expect researchers to use their method, they should provide them with an uncertainty range for $R_k$ as a function of the Angstrom exponent and $\kappa_{sca}$. Probably, higher errors are expected at higher $\kappa_{sca}$ values? This is certainly needed if they expect people to use the look-up-table. In general, the manuscript lacks of an appropriate treatment of errors despite the large expected errors for the hygroscopicity parameters.

**Response**: Thanks for your comment. We agree with the referee. The uncertainty range of $R_k$ based on the simulative results is shown in Fig.6b. The results is consistent with the referee's point that higher errors are expected at higher $\kappa_{sca}$ values. The maximum $\kappa_{sca}$ of the look up table is 0.4, if $R_k$ is 0.8 (close to the simulated highest $R_k$ shown in Fig.5b), the corresponding f(80%) is 2.6. According to the review of Titos et al. (2016), most of f(80%) for continental aerosols are lower than 2.6. This look up table already covers most situations for continental aerosol types.

[revised manuscript text omitted]

Traditionally, the Köhler theory (Petters and Kreidenweis, 2007) is widely used to describe the hygroscopic growth of aerosol particles and successfully used in laboratory studies for single component and some multicomponent particles. ~~However, it is found that most atmospheric aerosol particles usually consist of both organic and inorganic constituents (Murphy et al., 1998) rather than consist of a single component. Given this, a modified version of Köhler theory called κ-Köhler theory is proposed by Petters and Kreidenweis (2007) and widely used in recent ten years to study the hygroscopic growth of aerosol particles. The formula of this theory is expressed as the following~~In order to account for the mixed organic and inorganic composition of ambient aerosol, Petters and Kreidenweis (2007) proposed a modified version of Köhler theory called κ-Köhler theory to describe a single aerosol hygroscopic growth parameter, κ. The κ-Köhler equation, expressed in terms of the diameter growth factor, g(RH), is given in equation (1) below:

$$\frac{RH}{100}S = \frac{g D^3 - 1 D_d^3}{g D^3 - D_d^3(1-\kappa)} \cdot \exp\left(\frac{4\sigma_{s/a}\cdot M_{water}}{R\cdot T\cdot D_{dp}\cdot g\cdot \rho_w}\right) \qquad (1)$$

[revised manuscript text omitted]

~~In this paper, with measurements from a field campaign on the North China Plain (NCP), we first derived $\kappa$ values from measurements of $f(RH)$ with the traditional method and then compared them with the $\kappa$ values derived from High Humidity Tandem Differential Mobility Analyzer (HH-TDMA). HH-TDMA is a system very similar with HTDMA but is capable of operating at higher RH points (Liu et al., 2011). The relationships between $\kappa$ values derived from $f(RH)$ measurements and parameters used to fit measured $f(RH)$ curves are further examined and analyzed. Finally, basing on finished analysis about the relationship between $\kappa$ and $f(RH)$ fitting parameters, a novel method to directly derive the aerosol hygroscopicity parameter $\kappa$ based only on measurements from a humidified nephelometer system is proposed. This newly proposed approach makes it more convenient and cheaper for researchers to conduct aerosol hygroscopicity research with measurements of $f(RH)$.~~

**2. Site description and instruments**

~~In this study, the main part of used datasets is from the field campaign conducted at Wangdu (38°40′N, 115°08′E) during summer on the North China Plain (NCP). This field campaign was jointly conducted by Peking University, China and Leibniz Institute for Tropospheric Research, Germany. Wangdu site is located in the suburban district of Wangdu County, Hebei Province, China and situated adjacent to farmland and residential areas, it belongs to the typical region of the NCP. This observation~~

campaign lasted for about one month from 4 June, 2014 to 14 July, 2014. The measured $f(\mathrm{RH},\lambda)$
dataset was available from June 21[st], 2014, to July 1[st], 2014.

For datasets from Wangdu campaign. The chemical compositions of the aerosol particles with an
aerodynamic diameter of less than 2.5 μm (PM2.5) were analyzed based on the samples collected on
quartz and Teflon filters. Other instruments share one inlet which is placed on the roof of the container.
Regarding this inlet system, aerosol particles first entered an impactor which selected the aerosol
particles with an aerodynamic diameter of less than 10 μm, and then passed through a dryer which is
capable of reducing the RH of the sample air to lower than 30 %. In succession, the sample air passed
through a splitter and was allotted to different instruments according to their required flow rates. The
PNSD at dry state ranging from 3nm to 10μm was observed jointly by a Twin Differential Mobility
Particle Sizer (TDMPS, Leibniz Institute for Tropospheric Research (IfT), Germany; Birmili et al.
(1999)) and an Aerodynamic Particle Sizer (APS, TSI Inc., Model 3321) with a temporal resolution of
10 minutes. The absorption coefficient at 637 nm was measured using a Multi-angle Absorption
Photometer (MAAP Model 5012, Thermo, Inc., Waltham, MA USA) with a temporal resolution of 1
minute, and further used to calculate the mass concentrations of black carbon (BC) with a constant
mass absorption efficiency (MAE) of 6.6 $\mathrm{m^2g^{-1}}$. The growth factors of aerosol particles at six
selected particle diameters (30 nm, 50 nm, 100 nm, 150nm, 200 nm and 250 nm) at 98% RH condition
were obtained from the measurements of the HH-TDMA (Leibniz Institute for Tropospheric Research
(IfT), Germany; Hennig et al. (2005)). The $f(\mathrm{RH},\lambda)$ curves of aerosol particles with RH ranging from
about 50% to 90% were measured by a humidified nephelometer system which consists of two three-
wavelength integrating nephelometers (TSI Inc., Model 3563) and a humidifier. The humidifier was
used to moisten the air which will be sampled into the second nephelometer. Details of this humidified
nephelometer system please refer to (Kuang et al., 2016a).

PNSDs at dry state and mass concentrations of BC derived from MAAP measurements measured
at both Wuqing from 12 July to 14 August in 2009 and Xianghe from 9 July to 8 August in 2013 are
also used in this study to examine the influence of PNSD and BC on derivation of $\kappa$ values from
$f(\mathrm{RH})$ measurements and other relationships. Additionally, $\sigma_{sp}$ values which were observed during
these three field campaigns introduced before with a three-wavelength integrating nephelometer (TSI
Inc., Model 3563) are also used in Sect.4.3. Both Wuqing and Xianghe are representative regional
background sites of the NCP and locates in the northern part of the NCP. Details about these two

[revised manuscript text omitted]
^{-1}$. ~~The aerosol chemical compositions also change a lot during the observation period(Kuang et al., 2016a). The relative contributions of mass concentrations of organic matter to total PM2.5 mass concentrations range from 2% to 42%. Moreover, the relative contributions of mass concentrations of sulfate, nitrate and ammonium to total PM2.5 mass concentrations range from 5 to 50 %, 2 to 27 % and 1 to 21 %, respectively (Kuang et al., 2016a). These results imply that during this observation period, the aerosol hygroscopicity changes a lot whereafter corroborated by $f(\text{RH})$ measurements. Overall, $f(80\%)$ values range between 1.1 and 2.3 with an average of 1.8. Periods when deliquescent phenomena occur, $f(80\%)$ values are relatively higher with a variation range of 1.7 to 2.3 and their average is 2.0. This is because of the dominance of ammonium sulfate during periods when deliquescent phenomena occur. More detailed analysis about the frequently observed deliquescent phenomena during this field campaign please refer to (Kuang et al., 2016a). Furthermore, $\kappa$ values derived from $f(\text{RH})$ measurements by combining measurements of PNSD at dry state and mass concentrations of BCDuring deliquescent phenomena periods, $f(\text{RH})$ jumps when sample RH in the cavity of the~~

[revised manuscript text omitted]

fit observed $f$(RH) cycles well with $\gamma$ Method performs slightly better which is contrary to the results introduced by Brock et al. (2016), their results demonstrate that $\kappa_{sca}$ Method can better describe observed $f$(RH) than $\gamma$ Method. That is to say, $f$(RH) curves observed at different places or time periods may require different parameterization schemes to fit them best, however, in general both $\gamma$ Method and $\kappa_{sca}$ Method are good approaches to fit observed $f$(RH) curves.

Concerning $\gamma$ Method, previous studies usually examine the relationship between $\gamma$ and aerosol chemical compositions and established several parameterization schemes to fit $\gamma$ with mass fractions of different aerosol chemical compositions, including organic materials, sulfate and nitrate (Quinn et al., 2005;Titos et al., 2014;Zhang et al., 2015). However, to obtain a reliable estimation of $\gamma$, complete information of aerosol chemical compositions may be required which is difficult to get, and it is also hard to find a comprehensive description of $\gamma$ based on those complicated chemical compositions.

Single aerosol hygroscopicity parameter $\kappa$ can represent overall hygroscopicity of aerosol particles which contains influences of different chemical compositions on aerosol hygroscopicity, therefore may be used to better fit $\gamma$. In view of this, tThe relationship between $\kappa_{f(\mathrm{RH})}$ and $\gamma$ is investigated and shown in Fig.4aC. It is found that a pretty good an approximately linear relationship exists (square of correlation coefficient is 0.905) between $\kappa_{f(\mathrm{RH})}$ and $\gamma$, especially when $\kappa_{f(\mathrm{RH})}$ is larger than 0.215.

This correlation is far better than previously found relationships between $\gamma$ and aerosol chemical compositions (Quinn et al., 2005;Titos et al., 2014;Zhang et al., 2015) and statistical parameters which can be used to parameterize $\gamma$ with $\
[revised manuscript text omitted]
 of  two different campaigns. ~~Results show that great consistency is achieved between predicted and measured $R_\kappa$ values ( 92% points have relative difference less than 6%). Given this, the linkage between $\kappa_{f(\text{RH})}$ and $\kappa_{sca}$ is directly established and further used to estimate $\kappa_{f(\text{RH})}$. The comparison results demonstrate a pretty good agreement is achieved, all points lie nearby 1:1 line. The average absolute difference between $\kappa_{f(\text{RH})}$ derived from newly proposed method and traditional method is 0.005 and the square of correlation coefficient between them is 0.99.~~ The verification results demonstrate that a quite good estimation of $\kappa_{f(\text{RH})}$ can be achieved by using only measurements from a humidified nephelometer system, and this method is applicable at different sites and in different seasons. This newly proposed novel approach allow researchers to estimate $\kappa_{f(\text{RH})}$ without any additional information about PNSD and BC. This new finding directly links $\kappa$ and $f(\text{RH})$ and will make the humidified nephelometer system more convenient when it comes to aerosol hygroscopicity research. Finally, findings in this research may facilitate the intercomparison of aerosol hygroscopicity derived from different techniques, help for parameterizing $f(\text{RH})$ and predicting CCN properties with optical measurements.

**Acknowledgments**

[revised manuscript text omitted]

678 Ma, N., Zhao, C., Tao, J., Wu, Z., Kecorius, S., Wang, Z., Größ, J., Liu, H., Bian, Y., Kuang, Y., Teich, M., Spindler, G., Müller,
679 K., van Pinxteren, D., Herrmann, H., Hu, M., and Wiedensohler, A.: Variation of CCN activity during new particle formation
680 events in the North China Plain, Atmos. Chem. Phys., 16, 8593-8607, 10.5194/acp-16-8593-2016, 2016.

681 Murphy, D. M., Thomson, D. S., and Mahoney, M. J.: In Situ Measurements of Organics, Meteoritic Material, Mercury, and
682 Other Elements in Aerosols at 5 to 19 Kilometers, Science, 282, 1664-1669, 10.1126/science.282.5394.1664, 1998.

683 Petters, M. D., and Kreidenweis, S. M.: A single parameter representation of hygroscopic growth and cloud condensation
684 nucleus activity, Atmospheric Chemistry and Physics, 7, 1961-1971, 2007.

685 Petters, M. D., Carrico, C. M., Kreidenweis, S. M., Prenni, A. J., DeMott, P. J., Collett, J. L., and Moosmüller, H.: Cloud
686 condensation nucleation activity of biomass burning aerosol, Journal of Geophysical Research: Atmospheres, 114, n/a-
687 n/a, 10.1029/2009JD012353, 2009.

688 Pinnick, R. G., Jennings, S. G., and Chýlek, P.: Relationships between extinction, absorption, backscattering, and mass
689 content of sulfuric acid aerosols, Journal of Geophysical Research: Oceans, 85, 4059-4066, 10.1029/JC085iC07p04059,
690 1980.

691 Quinn, P. K., Bates, T. S., Baynard, T., Clarke, A. D., Onasch, T. B., Wang, W., Rood, M. J., Andrews, E., Allan, J., Carrico, C.
692 M., Coffman, D., and Worsnop, D.: Impact of particulate organic matter on the relative humidity dependence of light
693 scattering: A simplified parameterization, Geophys. Res. Lett., 32, n/a-n/a, 10.1029/2005GL024322, 2005.

694 Rose, D., Nowak, A., Achtert, P., Wiedensohler, A., Hu, M., Shao, M., Zhang, Y., Andreae, M. O., and Pöschl, U.: Cloud
695 condensation nuclei in polluted air and biomass burning smoke near the mega-city Guangzhou, China – Part 1: Size-
696 resolved measurements and implications for the modeling of aerosol particle hygroscopicity and CCN activity, Atmos.
697 Chem. Phys., 10, 3365-3383, 10.5194/acp-10-3365-2010, 2010.

698 Seinfeld, J. H., and Pandis, S. N.: Atmospheric chemistry and physics: from air pollution to climate change, John Wiley &
699 Sons, 2006.

700 Su, H., Rose, D., Cheng, Y. F., Gunthe, S. S., Massling, A., Stock, M., Wiedensohler, A., Andreae, M. O., and Pöschl, U.:

Hygroscopicity distribution concept for measurement data analysis and modeling of aerosol particle mixing state with
regard to hygroscopic growth and CCN activation, Atmos. Chem. Phys., 10, 7489-7503, 10.5194/acp-10-7489-2010, 2010.
Tao, J. C., Zhao, C. S., Ma, N., and Liu, P. F.: The impact of aerosol hygroscopic growth on the single-scattering albedo and
its application on the NO2 photolysis rate coefficient, Atmos. Chem. Phys., 14, 12055-12067, 10.5194/acp-14-12055-2014,
2014.
Titos, G., Lyamani, H., Cazorla, A., Sorribas, M., Foyo-Moreno, I., Wiedensohler, A., and Alados-Arboledas, L.: Study of the
relative humidity dependence of aerosol light-scattering in southern Spain, Tellus Ser. B-Chem. Phys. Meteorol., 66, 15,
10.3402/tellusb.v66.24536, 2014.
Titos, G., Cazorla, A., Zieger, P., Andrews, E., Lyamani, H., Granados-Muñoz, M. J., Olmo, F. J., and Alados-Arboledas, L.:
Effect of hygroscopic growth on the aerosol light-scattering coefficient: A review of measurements, techniques and error
sources, Atmospheric Environment, 141, 494-507, http://dx.doi.org/10.1016/j.atmosenv.2016.07.021, 2016.
Wex, H., Neususs, C., Wendisch, M., Stratmann, F., Koziar, C., Keil, A., Wiedensohler, A., and Ebert, M.: Particle scattering,
backscattering, and absorption coefficients: An in situ closure and sensitivity study, Journal of Geophysical Research-
Atmospheres, 107, 18, 10.1029/2000jd000234, 2002.
Wu, Z. J., Zheng, J., Shang, D. J., Du, Z. F., Wu, Y. S., Zeng, L. M., Wiedensohler, A., and Hu, M.: Particle hygroscopicity and
its link to chemical composition in the urban atmosphere of Beijing, China, during summertime, Atmos. Chem. Phys., 16,
1123-1138, 10.5194/acp-16-1123-2016, 2016.
You, Y., Smith, M. L., Song, M., Martin, S. T., and Bertram, A. K.: Liquid–liquid phase separation in atmospherically relevant
particles consisting of organic species and inorganic salts, International Reviews in Physical Chemistry, 33, 43-77,
10.1080/0144235X.2014.890786, 2014.
Zhang, L., Sun, J. Y., Shen, X. J., Zhang, Y. M., Che, H., Ma, Q. L., Zhang, Y. W., Zhang, X. Y., and Ogren, J. A.: Observations of
relative humidity effects on aerosol light scattering in the Yangtze River Delta of China, Atmos. Chem. Phys., 15, 8439-
8454, 10.5194/acp-15-8439-2015, 2015.
Zhao, C., Tie, X., and Lin, Y.: A possible positive feedback of reduction of precipitation and increase in aerosols over eastern
central China, Geophys. Res. Lett., 33, L11814, 10.1029/2006GL025959, 2006.
Zieger, P., Fierz-Schmidhauser, R., Weingartner, E., and Baltensperger, U.: Effects of relative humidity on aerosol light
scattering: results from different European sites, Atmos. Chem. Phys., 13, 10609-10631, 10.5194/acp-13-10609-2013,
2013.
Zieger, P., Fierz-Schmidhauser, R., Poulain, L., Muller, T., Birmili, W., Spindler, G., Wiedensohler, A., Baltensperger, U., and
Weingartner, E.: Influence of water uptake on the aerosol particle light scattering coefficients of the Central European
aerosol, Tellus Ser. B-Chem. Phys. Meteorol., 66, 10.3402/tellusb.v66.22716, 2014.

**Table 1**. Locations, time periods and used datasets of five field campaigns

[revised manuscript text omitted]

---

## Author Response (AR2)

**Response to reviewer #2**

Thanks for the reviewer's helpful suggestions! The comments are addressed point-by-point and responses are listed below.

**Comment:** In my previous review, I suggested the authors to improve the English grammar and the organization of the manuscript (especially the description of the methods). They have worked on that, but not enough from my point of view. I suggest to introduce sub-sections for each method. In addition, apart from methods 1-4, they introduced γ-method and κsca-method. What is the difference between method 3 and κsca-method?

In line 156, the authors mentioned four campaigns, but Table 1 listed up to five campaigns. It is not clear to me if the data from Wuqing and Xianghe campaigns are used or not, and for what? Why the data is not corrected? To which corrections do the authors refer?

**Response**: Thanks for your comment. In the revised manuscript, each method is introduced in a sub-section. The method 3 is same with κsca-method, we have changed the term "κsca-method" to Method 3.

In this paper, datasets from five field campaigns are used. The following sentence is included in Sect.2 of the manuscript: "Dataset includes aerosol PNSDs at dry state, mass concentrations of BC and $\sigma_{sp}$ values of different wavelengths from the following four campaigns which are listed in Table 1 are referred to as dataset D1: two campaigns conducted in Wuqing, Xianghe campaign, Wangdu campaign before 21 June, 2014".

Dataset D1 are used for simulating the look up table shown in Fig.6a of the manuscript. This information and the reason of using datasets from four field campaigns is included in the manuscript, and expressed as the following: "To better understand the relationship between $\kappa_{f(\mathrm{RH})}$ and $\kappa_{sca}$, all PNSDs at dry state (shown in Fig.3a) along with mass concentrations of BC from dataset D1 are used to simulate the relationship between $\kappa_{f(\mathrm{RH})}$ and $\kappa_{sca}$ with Mie and κ-Köhler theories. The aim of including PNSD and BC information from different campaigns is to simulate variations of $R_k$ under different conditions". The reason that datasets from Gucheng campaign and Wangdu campaign during the period from 21 June, 2014, to 1 July , 2014 (the period when measurements from the humidified nephelometer system were available) are not used for producing the look up table is to make sure that the verification datasets are totally independent of the look up table.

In the revised manuscript, the type and reason about the correction is added and expressed as the following: "Note that measured $\sigma_{sp}$ values of dataset D1 are not corrected for angular truncation errors. This is because that dataset D1 is used for producing the look up table of the newly proposed method, and it is expected that the Ångström exponent calculated from measured $\sigma_{sp}$ values can be directly used as input for the newly proposed method".

**Comment**: One of my main concerns in my previous review was related with the error/uncertainties estimates of using the "new method" proposed in the manuscript. Although they have included a new dataset for testing the method, they do not quantify

the errors of using this method to estimate the hygroscopicity parameter. They should address this point if they expect their method to be useful for the scientific community.

**Response**: Thanks for your comment. We agree with the reviewer, it is best that we can fully quantify the uncertainties of using this method. Except for the measurement uncertainty of f(RH), uncertainties of this new method are arisen from the look up table shown in Fig.6a of the manuscript. With regard to uncertainties of the look up table, we can not conduct a thorough uncertainty analysis because that we can not know all possible conditions (PNSD, BC, and other factors). In processes of producing the look up table, we included as many datasets as possible from different field campaigns (conducted in different sites and seasons) for simulating this look up table to cover different PNSD and BC conditions as much as possible. In this light, we think uncertainty ranges of $R_k$ under different conditions shown in Fig.6b which are based on the simulative results can be treated as the uncertainty analysis of this look up table to some extent.

**Comment**: Furthermore, the authors need to make clear that the method is valid for NCP aerosols. They stated in the abstract that the method is "applicable in different sites and seasons" but they have just tested the method in another site from the NCP (probably with similar aerosol characteristics…)

**Response**: Thanks for your comment. The following discussion is added in the revised manuscript: "The processes of simulating the look up table are independent of the size-resolved $\kappa$ distribution, and used PNSDs are from four different field campaigns which were conducted in different sites and seasons of the NCP. The verification datasets from

two different field campaigns are totally independent of the look up table and from different sites and seasons of the NCP. These results demonstrate that the newly proposed method is applicable in different sites and seasons of the NCP. The results shown in Fig.6b demonstrate that if Ångström exponent and $\kappa_{sca}$ are fixed, then $R_k$ varies little. The maximum $\kappa_{sca}$ of the look up table is 0.4, if $R_k$ is 0.8 (close to the simulated highest $R_k$ shown in Fig.5b), the corresponding $f(\text{RH} = 80\%)$ is 2.6. According to the review of Titos et al. (2016), most of $f(\text{RH} = 80\%)$ values for continental aerosols are lower than 2.6. The Ångström exponent range of the look up table is 0.4 to 2.0. Which demonstrate that the look up table shown in Fig.6a already covers large variation ranges of Ångström exponent and $\kappa_{sca}$ and can be used under different conditions. Thus, the newly proposed method of deriving $\kappa_{f(\text{RH})}$ might be also applicable in other regions around the world".

Titos, G., Cazorla, A., Zieger, P., Andrews, E., Lyamani, H., Granados-Muñoz, M. J., Olmo, F. J., and Alados-Arboledas, L.: Effect of hygroscopic growth on the aerosol light-scattering coefficient: A review of measurements, techniques and error sources, Atmospheric Environment, 141, 494-507, http://dx.doi.org/10.1016/j.atmosenv.2016.07.021, 2016.